# ESCRT-III drives the final stages of CUPS maturation for unconventional protein secretion

Amy J Curwin[1,2], Nathalie Brouwers[1,2], Manuel Alonso Y Adell[3], David Teis[3], Gabriele Turacchio[4], Seetharaman Parashuraman[4], Paolo Ronchi[5], Vivek Malhotra[1,2,6*]

[1]Centre for Genomic Regulation, Barcelona Institute of Science and Technology, Barcelona, Spain; [2]Universitat Pompeu Fabra, Barcelona, Spain; [3]Division of Cell Biology, Biocenter, Medical University of Innsbruck, Innsbruck, Austria; [4]Institute of Protein Biochemistry, National Research Council of Italy, Naples, Italy; [5]Electron Microscopy Core Facility, European Molecular Biology Laboratory, Heidelberg, Germany; [6]Institució Catalana de Recerca i Estudis Avançats (ICREA), Barcelona, Spain

**Abstract** The unconventional secretory pathway exports proteins that bypass the endoplasmic reticulum. In *Saccharomyces cerevisiae*, conditions that trigger Acb1 secretion via this pathway generate a Grh1 containing compartment composed of vesicles and tubules surrounded by a cup-shaped membrane and collectively called CUPS. Here we report a quantitative assay for Acb1 secretion that reveals requirements for ESCRT-I, -II, and -III but, surprisingly, without the involvement of the Vps4 AAA-ATPase. The major ESCRT-III subunit Snf7 localizes transiently to CUPS and this was accelerated in *vps4Δ* cells, correlating with increased Acb1 secretion. Microscopic analysis suggests that, instead of forming intraluminal vesicles with the help of Vps4, ESCRT-III/Snf7 promotes direct engulfment of preexisting Grh1 containing vesicles and tubules into a saccule to generate a mature Acb1 containing compartment. This novel multivesicular / multilamellar compartment, we suggest represents the stable secretory form of CUPS that is competent for the release of Acb1 to cells exterior.

*For correspondence: vivek.malhotra@crg.eu

## Introduction

A large number of signal sequence lacking proteins are secreted by eukaryotic cells and the examples include: Acyl CoA binding protein 1 (Acb1), also known as diazepam binding inhibitor (DBI) that acts as an allosteric effectors of GABA ionotrophic receptor (*Costa and Guidotti, 1991*; *Gandolfo et al., 2001*); FGF1and FGF2 that are required for angiogenesis (*Jackson et al., 1992*; *Schäfer et al., 2004*); the β-galactoside–specific lectins galectin 1 and 3, blood coagulation factor XIIIa, macrophage migration inhibitory factor (MIF), interleukin (IL)-1ß, and the engrailed homeoprotein (*Dupont et al., 2011*; *Flieger et al., 2003*; *Grundmann et al., 1988*; *Joliot et al., 1997*; *Rubartelli et al., 1990*; *Lutomski et al., 1997*; *Menon and Hughes, 1999*; *Manjithaya et al., 2010*; *Duran et al., 2010*; *Kinseth et al., 2007*; *Nickel and Seedorf, 2008*; *Rabouille et al., 2012*; *Subramani and Malhotra, 2013*; *Zang et al., 2015*). How are these proteins that cannot enter the endoplasmic reticulum - Golgi complex pathway of protein secretion exported from cells? Is there a common pathway of their export? Does their release from the cytoplasm to the extracellular space involve a membrane bounded vesicular intermediate or are they translocated directly from the cytoplasm to the extracellular space via a translocator? Progress in our understanding of these issues of

**eLife digest** Cells produce thousands of different proteins with a variety of different roles in the body. Some proteins, for example the hormone insulin, perform roles outside of the cell and are released from cells in a process that has several stages. In the first step, newly-made insulin and many other "secretory" proteins enter a compartment called the endoplasmic reticulum. Once inside, these proteins can then be loaded into other compartments and transported to the edge of the cell.

There is another class of secretory proteins that are released from the cell without first entering the endoplasmic reticulum, in a process termed "unconventional protein secretion". A protein called Acb1 is released from yeast cells in this manner. Previous research identified a compartment that might be involved in this process. However, it is not clear how this compartment (named CUPS) forms, and what role it plays in unconventional protein secretion.

Curwin et al. investigated how CUPS form in yeast cells, and whether the compartment contains Acb1 proteins. The experiments reveal that after CUPS form they need to mature into a form that is involved in the release of Acb1 proteins from the cell. This maturation process involves some, but not all, of the same genes as those involved in producing another type of compartment in cells called a multivesicular body. Acb1 is only found in the mature CUPS and multivesicular bodies are not involved in the release of this protein from the cell.

Curwin et al.'s findings shed some light on how Acb1 and other secretory proteins can be released from cells without involving the endoplasmic reticulum. Future challenges are to reveal how CUPS capture cargo and find out how Acb1 leaves the CUPS to exit the cell.

fundamental importance has been slow because these proteins are released in exceptionally small quantities in a cell type specific and a signal dependent manner (*Malhotra, 2013*). This class of proteins is typically detected in the extracellular space by a functional bioassay or ELISA, but these assays do not distinguish between processes involved in the secretion of full-length proteins from those that process the respective cargoes for their presentation in a functional active form. These technical limitations have hampered our understanding of the mechanism of unconventional protein secretion.

In 2007, we made a surprising finding that an ER exit site and Golgi membrane associated peripheral protein GrhA was required for nutrient starvation induced secretion of signal sequence lacking Acyl CoA binding protein (AcbA) of 87 amino acids by *Dictyostelium discoideum* (*Kinseth et al., 2007*). In the extracellular space, AcbA is proteolytically processed into a 34 amino acid peptide, SDF-2 (spore differentiation factor 2). SDF-2 is required for rapid encapsulation of the prespore cells (*Anjard and Loomis, 2005*; *Anjard et al., 1998*). Subsequent analysis revealed that secretion of AcbA orthologs of the yeasts *Saccharaomyces cerevisiae* and *Pichia pastoris* also required GrhA ortholog called Grh1 (*Duran et al., 2010*; *Manjithaya et al., 2010*). In addition, a subset of genes that ordinarily function in the biogenesis of multi-vesicular body (MVB), targeting of membranes to endosomes, fusion of membranes with the plasma membrane, and autophagosome formation were also required for Acb1 secretion (*Duran et al., 2010*; *Manjithaya et al., 2010*). However, the secretion of Acb1 was measured by an assay that detected the activity of SDF-2 or an SDF-2-like peptide. This procedure does not distinguish proteins required directly for Acb1 secretion from those with a role in its modification or processing to generate a functional SDF-2. In our subsequent analyses, we discovered that Grh1, upon incubation of yeast in starvation medium, translocated from its normal ER exit site/early Golgi residence to one or two larger membrane bound compartments. Based on the shape of the membranes containing Grh1, we have called these compartments CUPS (Compartment for Unconventional Protein Secretion) (*Bruns et al., 2011*). In addition to Grh1, CUPS contain the early Golgi components Bug1, Uso1 and Sed5, but form independent of COPII and COPI dependent vesicular transport (*Cruz-Garcia et al., 2014*). The biogenesis of CUPS requires the PI 4-kinase Pik1 and the Arf-GEF Sec7. Interestingly, in a *vps34Δ* mutant CUPS form but breakdown indicating the requirement of PI3P production by Vps34 in the stability of the CUPS (*Bruns et al., 2011*; *Cruz-Garcia et al., 2014*).

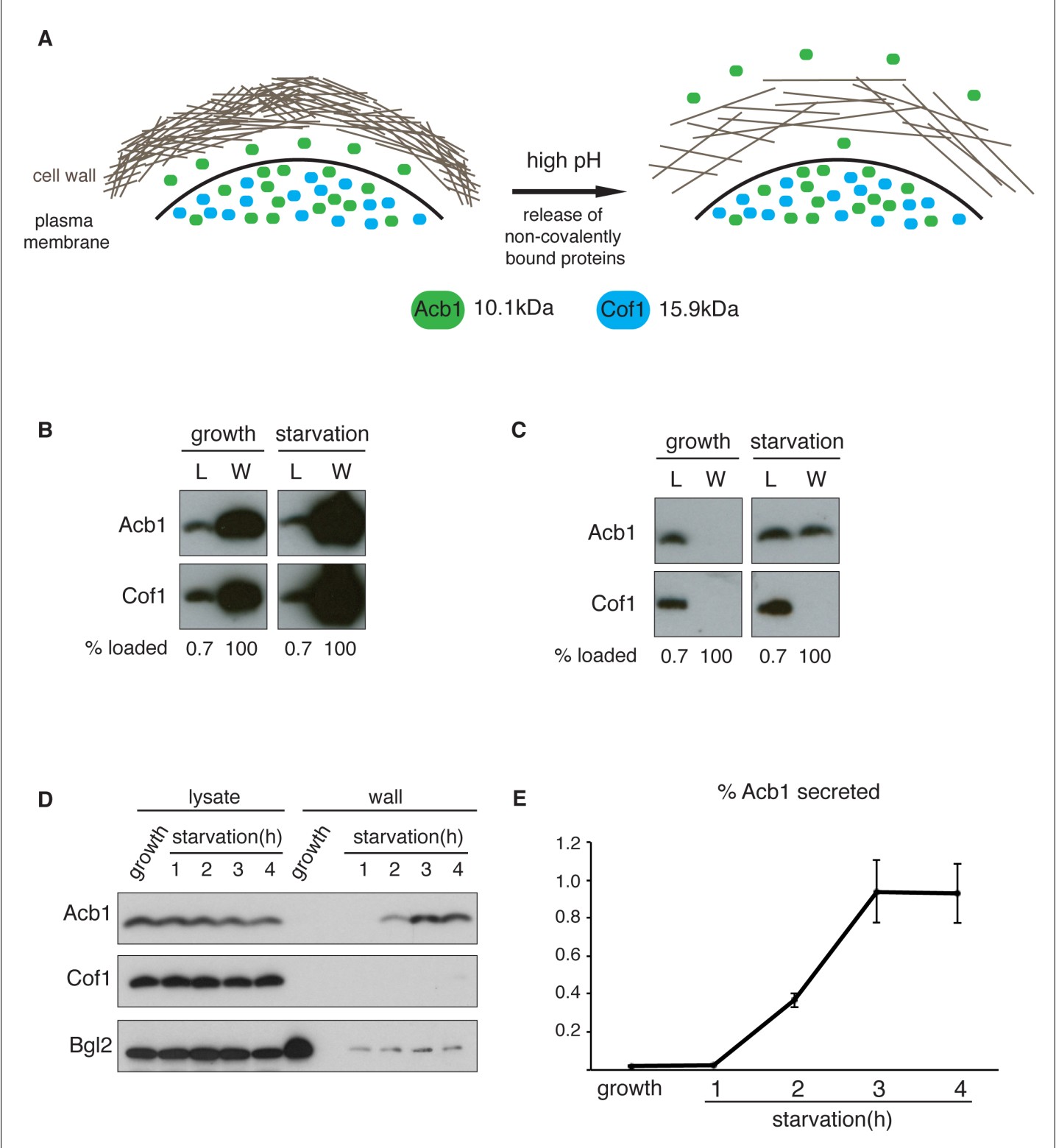

**Figure 1.** A quantitative assay for Acb1 secretion. (**A**) The cell wall is a highly-charged, porous meshwork of glucans, chitin, and mannoproteins. Incubation in high pH buffers loosens the cell wall, thus allowing some non-covalently bound proteins to be released. (**B**) Standard cell wall extraction procedures employed thus far cause cell lysis. Wild type cells were grown to mid-logarithmic phase, washed twice, and cultured in 2% potassium acetate for 2.5 hr (starvation). Cell wall proteins were extracted from equal numbers of growing and starved cells in 100 mM Tris-HCl pH 9.4, 10 mM DTT with mixing at 350 rpm for 30 min at 37°C, followed by precipitation with TCA. Lysates (L) and cell wall-extracted proteins (W) were analyzed by

*Figure 1 continued on next page*

*Figure 1 continued*

western blot. (C) Mild cell wall extraction conditions do not cause lysis and reveal starvation-specific release of Acb1. Wild type cells were grown to mid-logarithmic phase, washed twice, and incubated in 2% potassium acetate for 2.5 hr (starvation). Cell wall proteins were extracted from equal numbers of growing and starved cells in 100 mM Tris-HCl pH 9.4, 2% sorbitol for 10 min on ice followed by precipitation with TCA. Lysates (L) and cell wall-extracted proteins (W) were analyzed by western blot. (D) Time course of Acb1 secretion during starvation. Wild type cells were grown to mid-logarithmic phase, washed twice, and cultured in 2% potassium acetate for the indicated times. Cell wall proteins were extracted in the mild conditions described in (C) and analyzed by western blot. (E) The ratio of wall/lysate Acb1 was determined and the average amount of secreted Acb1 at 3 hr was calculated to be 0.94% (SEM = 0.16%) and at 4 hr 0.93% (SEM = 0.16%) of total cellular Acb1 (n = 3).

We have now developed a procedure to measure full length secreted Acb1 by extracting the yeast cell wall without causing cell lysis. We have used this assay to characterize the role of the ESCRT proteins in CUPS biogenesis and Acb1 secretion. Our findings reveal that ESCRT-I, -II and –III are involved in Acb1 secretion. In contrast neither ESCRT-0 nor Vps4 are required for this process. These results indicate a Vps4 independent role of ESCRT-III in membrane remodeling. We present the ultra structural analysis of CUPS and the findings that Snf7, the ESCRT-III component, attaches to CUPS during maturation and is required for their stability. The stable CUPS are found to contain Acb1. The description and the significance of our findings follow.

## Results

### A quantitative assay for Acb1 secretion

We were unable to detect full-length Acb1 or SDF-2 directly in the medium of starving *S.cerevisiae* by immunoprecipitation, western blotting and mass spectrometry (data not shown). We reasoned that full-length Acb1 was likely secreted into the periplasmic space that is between plasma membrane and the cell wall and this pool was cleaved to generate SDF-2. Once processed, SDF-2 could diffuse into the medium because of its small size (34 amino acids) and/or charge. The cell wall of yeast is composed of glucans, chitin and an outer layer of highly negatively-charged mannoproteins (*Lipke and Ovalle, 1998*). Incubating cells in alkaline buffer loosens the cell wall and releases a population of non-covalently bound cell wall proteins (*Figure 1A*) (*Klis et al., 2007*; *Mrsă et al., 1997*). In fact, this procedure has been used to report the secretion of signal sequence lacking gluconeogenic, glycolytic enzymes, and the exogenously expressed human Galectin-1 (*Cleves et al., 1996*; *Giardina et al., 2014*). But how much of these proteins are released as a result of cell lysis by this procedure?

To distinguish secreted Acb1 from that which leaks into the extracellular space due to cell lysis, we compared the presence of Acb1 in the extracellular space to cofilin (Cof1), which is not secreted. Acb1 and Cof1 are both small proteins of 10.1 kDa and 15.9 kDa, respectively, they have similar predicted isoelectric points, and are abundant cytosolic proteins estimated at 142817 and 201065 molecules/cell, respectively (*Kulak et al., 2014*). Cell leakage, rupture of the plasma membrane or lysis during the experimental procedures should have similar effects on Acb1 and Cof1.

Yeast were grown to mid-logarithmic phase and either left untreated or washed twice and starved of nitrogen and glucose by incubation in 2% potassium acetate (hereafter referred to as starvation). After 2.5 hr equal number of growing and starved cells were harvested. The cell wall was extracted by the standard procedure for removal of non-covalently bound proteins by incubation with a pH 9.4 Tris-HCl buffer in combination with reducing agents and mixing at 37°C (*Cleves et al., 1996*; *Giardina et al., 2014*). The proteins extracted under these conditions were analyzed by western blotting with anti-Acb1 and Cof1 antibodies, respectively. The results revealed high levels of Cof1 and Acb1 in the cell wall extracts of both growing and starved cells (*Figure 1B*), suggesting partial cell lysis. We therefore modified the procedure and incubated cells in pH 9.4 Tris-HCl buffer without reducing agents and in the presence of 2% sorbitol to protect against the osmotic stress resulting from loosening the cell wall. The mixture was kept on ice to further preserve the integrity of the plasma membrane. Under these conditions Acb1, but not Cof1, was detected in the cell wall extract of starving cells (*Figure 1C*). Next we monitored secretion of Acb1 during starvation over time comparing cell wall extracts of growing cells and cells starved for up to 4 hr. The spore viability assay previously indicated a burst in SDF-2 activity at 2.5 hr post-induction of starvation (*Duran et al.,*

2010). Consistently, secretion of full-length Acb1 into the cell wall/periplasm followed similar dynamics, peaking at 3 hr of starvation (*Figure 1D*). The levels of a soluble cell wall protein, Bgl2, were also analyzed as a marker of cell wall extraction efficiency. The amount of Bgl2 extracted from starved cells remained constant throughout starvation, indicating that secretion of Acb1 increased during starvation (*Figure 1D*). The levels of cell wall extracted and lysate Acb1 were quantitated and the amount of Acb1 secreted after 3 hr of starvation was calculated to be on average 1% of the total intracellular Acb1 (*Figure 1E*). This procedure to detect Acb1 is not without limitations however. Culturing cells at 37°C, prior to extraction, resulted in Cof1 release (data not shown), which precludes testing of temperature-sensitive yeast mutants for their involvement in Acb1 secretion. High concentrations of DMSO, which is often used as a solvent to dissolve chemicals such as Latrunculin A or Brefeldin A, also led to Cof1 release (data not shown). Therefore, DMSO soluble chemical inhibitors, in general, cannot be tested for their effects on Acb1 secretion. Finally, some yeast mutant strains are hyper osmosensitive and cannot therefore be tested by our procedure as they reveal a persistent Cof1 signal upon cell wall extraction. Regardless of these caveats, this assay for the first time provides a quantitative and a reliable measure of Acb1 secretion.

We first tested the involvement of Grh1 that has been shown previously to be involved in Acb1 secretion based on the spore viability assay in *D.dictyostelium* (*Kinseth et al., 2007*; *Duran et al., 2010*). Wild type and *grh1Δ* cells were grown to mid-logarithmic phase, starved in potassium acetate and extracted by the procedure described above. We chose to monitor secretion after 2.5 hr starvation to further avoid the potential of lysis associated with longer periods of starvation. The ratios of cell wall to lysate Acb1 from potassium acetate cultured cells were quantitated and the effect of loss of Grh1 was normalized to percent of wild type. No Acb1 was detected in the cell wall of growing *grh1Δ* cells (data not shown), while *grh1Δ* cells incubated in potassium acetate showed ∼80% reduction in Acb1 secretion compared to wild-type cells (*Figure 2A–B*). This confirms the involvement of Grh1 in Acb1 secretion.

## ESCRT-I, II and III are required for Acb1 secretion

Next, we tested the role of the endosomal sorting complexes required for transport (ESCRTs) that are involved in the formation of intraluminal vesicles in endosomes to generate a multivesicular body (MVB), some of which we previously reported to play a role in Acb1 secretion (*Duran et al., 2010*). Specifically, we tested Vps23, Vps37 and Vps28 of ESCRT-I, which showed 82%, 82% and 69% reduction in Acb1 secretion, respectively. All members of ESCRT-II, consisting of Vps36, Vps25 and Vps22, were tested resulting in 82%, 79% and 72% decrease in Acb1 secretion, respectively. Similarly, all four core components of ESCRT-III, Snf7, Vps20, Vps24 and Vps2, were examined and all showed similar reductions in secretion of 69%, 72%, 78% and 67% respectively (*Figure 2C–D*). Vps27 and Hse1 of ESCRT-0 were not required for Acb1 secretion (*Figure 2C–D*).

To complete the assessment of the role of the MVB pathway in Acb1 secretion we tested the role of Vps4, the AAA-ATPase required for the disassembly of ESCRT-III (*Babst et al., 1998*). Yet, loss of Vps4 did not affect Acb1 secretion (*Figure 2E–F*).

Vps4 activity in the MVB pathway is regulated by interactions with various accessory proteins. Did2 and Ist1 proteins modulate ESCRT-III disassembly by recruiting Vps4, while Vta1 and Vps60 form a complex that positively regulates the activity of Vps4 on endosomes (*Xiao et al., 2008*; *Azmi et al., 2006*; *Rue et al., 2008*; *Shestakova et al., 2010*; *Nickerson et al., 2006*). Deletion strains for each of these accessory proteins were tested for Acb1 secretion by the cell wall extraction assay and none exhibited a reduction in Acb1 secretion (*Figure 2E–F*).

These results suggest that Acb1 secretion requires ESCRT-III function but not Vps4 activity. This is an unexpected finding since all other known ESCRT dependent processes require ESCRT-III and Vps4.

## Role of ESCRT proteins in CUPS biogenesis and stability

Our previous findings revealed the requirement of a subset of ESCRT proteins in CUPS biogenesis (*Bruns et al., 2011*). In our previous analysis Vps27, Hse1, Vps23, Mvb12, Vps23, and Vps4 were found to have no role in CUPS biogenesis, whereas Vps36, Vps25, Vps2 and Vps20 deletions revealed varying, intermediate effects at 4 hr after starvation, which we know now to be much longer than required for Acb1 secretion or CUPS biogenesis (*Bruns et al., 2011*; *Cruz-Garcia et al., 2014*).

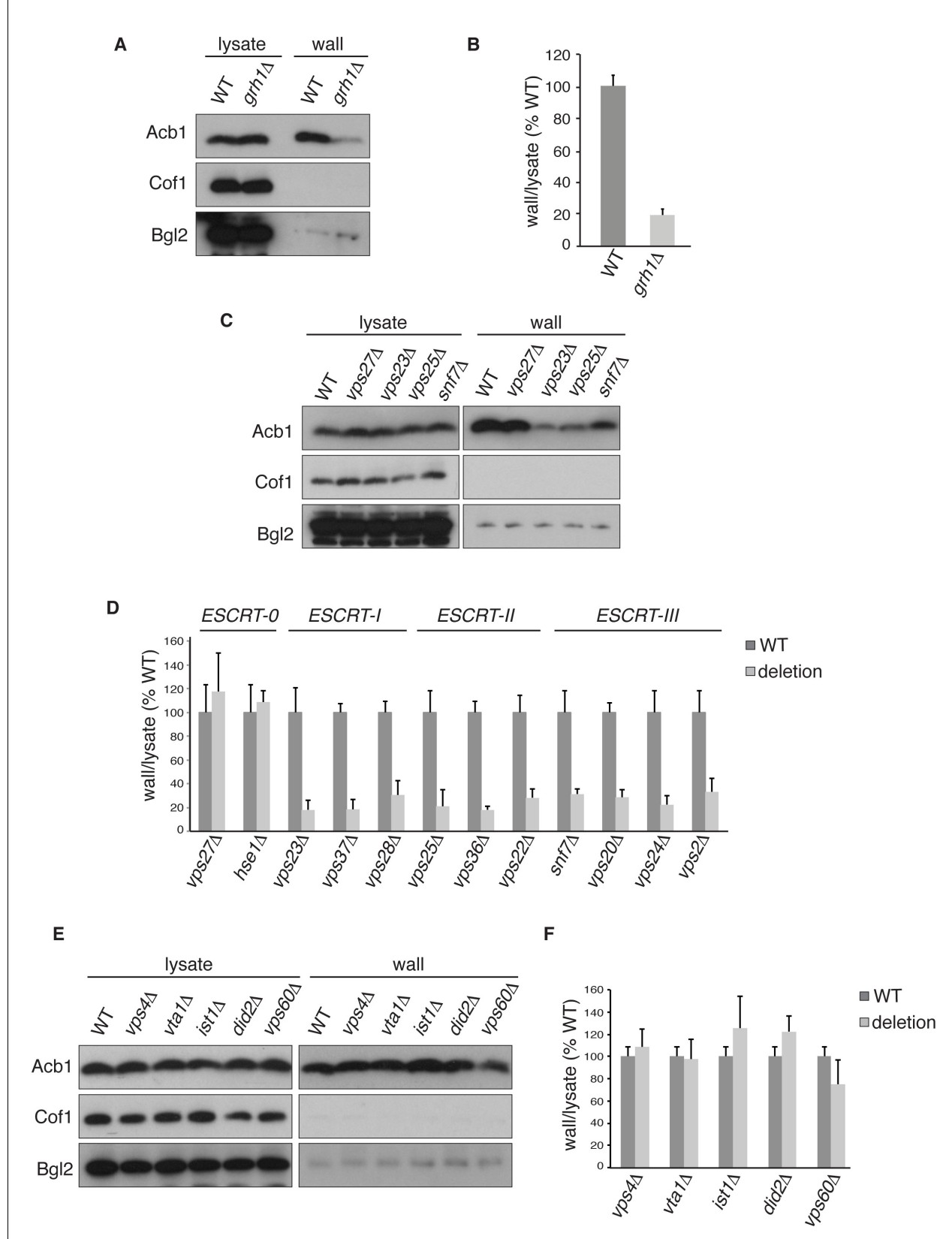

**Figure 2.** Acb1 secretion requires Grh1 and a subset of ESCRT proteins. Wild type and deletion strains were grown to mid-logarithmic phase, washed twice, and incubated in 2% potassium acetate for 2.5 hr. Cell wall proteins were extracted from equal numbers cells in 100 mM Tris-HCl pH 9.4, 2%
*Figure 2 continued on next page*

*Figure 2 continued*

sorbitol for 10 min on ice followed by precipitation with TCA. Lysates and cell wall-extracted proteins were analyzed by western blot. The ratio of wall/lysate Acb1 during starvation was determined and compared to that of wild type in each experiment. Statistical analyses were performed for each gene deletion and are represented as% of wild type (paired student's t-test). (A) Grh1 is required for Acb1 secretion. Representative cell wall extractions monitored by western blot of wild type and *grh1Δ* cells. (B) *grh1Δ* cells secreted on average 80.3% less Acb1 than wild-type cells (*p* = 0.0045, SEM = 3.8%, n= 4). (C) Acb1 secretion requires ESCRT complexes I, II and III but not ESCRT-0. Representative cell wall extractions monitored by western blot of wild type and one deletion strain from each ESCRT complex. (D) Quantification of all ESCRT deletion strains tested (n = 4 or more). ESCRT-0; *vps27Δ* (+17%, SEM = 3.2%, *p* > 0.2), *hse1Δ* (+8%, SEM = 0.9%, *p* > 0.2). ESCRT-I; *vps23Δ* (-81.8%, SEM = 7.9%, *p* = 0.0046), *vps37Δ* (-81.5%, SEM = 8.4%, *p* = 0.001), *vps28Δ* (-69.4%, SEM = 11.8%, *p* = 0.026). ESCRT-II; *vps25Δ* (-78.8%, SEM = 14.1%, *p* = 0.021), *vps36Δ* (-81.7%, SEM = 2.9%, *p* = 0.0055), *vps22Δ* (-71.6%, SEM = 7.7%, *p* = 0.032). ESCRT-III; *snf7Δ* (-68.5%, SEM = 4.4%, *p* = 0.055), *vps20Δ* (-71.5%, SEM = 6.5%, *p* = 0.022), *vps24Δ* (-77.4%, SEM = 7.5%, *p* = 0.049), *vps2Δ* (-66.7%, SEM = 11.3%, *p* = 0.0006). (E) Vps4 and accessory proteins are not required for Acb1 secretion. Representative cell wall extractions monitored by western blot of wild type and indicated deletion strains. (F) Quantification of indicated deletion strains (n = 4 or more). No differences were determined to be statistically significant.

In the current analysis we examined the requirements of ESCRT proteins in CUPS formation at 30 min and 2.5 hr of starvation to discern effects on biogenesis and stability and included ESCRT proteins that had not been tested so far. Cells were quantified for presence of CUPS (1–3 large punctae), intermediate CUPS formation (large punctae and small punctae) or no CUPS (only small punctae). None of the ESCRT proteins tested affected the initial CUPS formation and organization after 30min of starvation (*Figure 3A*). After 2.5 hr of starvation, we observed a requirement of ESCRT-II (Vps36, Vps22 and Vps25) and ESCRT-III complexes (Vps20, Vps2, Vps24 and Snf7), but not ESCRT-0 (Vps27 and Hse1), ESCRT-I (Vps23 and Vps28) or Vps4 proteins, as also reported previously after 4 hr starvation (*Figure 3A*) and *Bruns et al. (2011)*. The loss of Vps20 and Snf7 of ESCRT-III had the most dramatic effect on CUPS formation after 2.5 hr of starvation (*Figure 3A–B*).

We have shown previously that upon shifting cells from starvation to growth medium CUPS relocated to the ER by a Sly1 and COPI dependent pathway (*Cruz-Garcia et al., 2014*). We assessed whether ESCRT proteins affected CUPS absorption into the ER upon re-growth. The same cells as above were starved for 2.5 hr and then incubated in growth medium for 45 min. The wild type cells revealed the expected relocation of Grh1-2xGFP to ER exit sites/early Golgi membranes. This relocation was unaffected by loss of any ESCRT protein, including Vps4 (data not shown and *Figure 3C*). Therefore, CUPS maturation during starvation depends on ESCRT-II and III complex members, but CUPS reabsorption into the ER upon re-growth is independent of ESCRT proteins.

## Snf7 localizes transiently to CUPS and is required for CUPS stability

Loss of Snf7 exhibited a strong defect in CUPS morphology after 2.5 hr of starvation, and because it is the most abundant ESCRT-III protein, contributing 50% of the predicted 450 kDa ESCRT-III complex, we decided to localize Snf7 in starving cells (*Teis et al., 2008*). The C-terminal fusion of bulky tags such as GFP or RFP to all four ESCRT-III subunits is known to act as dominant-negative with respect to MVB sorting by interfering with auto inhibition of the C-terminal tail (*Teis et al., 2008*). We examined Grh1-2xGFP localization when Snf7-RFP was integrated in the genome and found it also acted as dominant-negative with respect to CUPS stability, exhibiting the same phenotype as loss of Snf7 (*Figure 4A*). Snf7-RFP localized to multiple punctate elements, representing dysfunctional MVBs. Interestingly, upon starvation, the Grh1 and Snf7 containing structures were often juxtaposed or slightly overlapping (*Figure 4A*). An established procedure to overcome this inhibitory effects is to co-express wild type and Snf7-RFP in the same cells (*Guizetti et al., 2011*). To ensure that co-expression of Snf7-RFP did not affect MVB pathway we assessed the trafficking of MVB cargo Cps1 in cells expressing empty vector or Snf7-RFP under the control of its own promoter. Our findings reveal that expression of Snf7-RFP did not block trafficking of GFP-Cps1 to the vacuole (*Figure 4—figure supplement 1*). We therefore co-expressed exogenous Snf7-GFP with untagged endogenous Snf7 in cells expressing Grh1-2xmCherry. Under these conditions CUPS formation after 2.5 hr of starvation was largely restored, with more than 70% of cells exhibiting morphological normal CUPS (*Figure 4B*). In growth conditions, Snf7-GFP localized to small punctate elements and the vacuole membrane (*Figure 4B*). The vacuole membrane location is likely a result of inefficient disassembly of tagged Snf7. Upon starvation, Snf7 localized predominantly to 1–3 large structures per cell, and the vacuole membrane labeling was largely reduced (*Figure 4B*). Cells with very high Snf7-

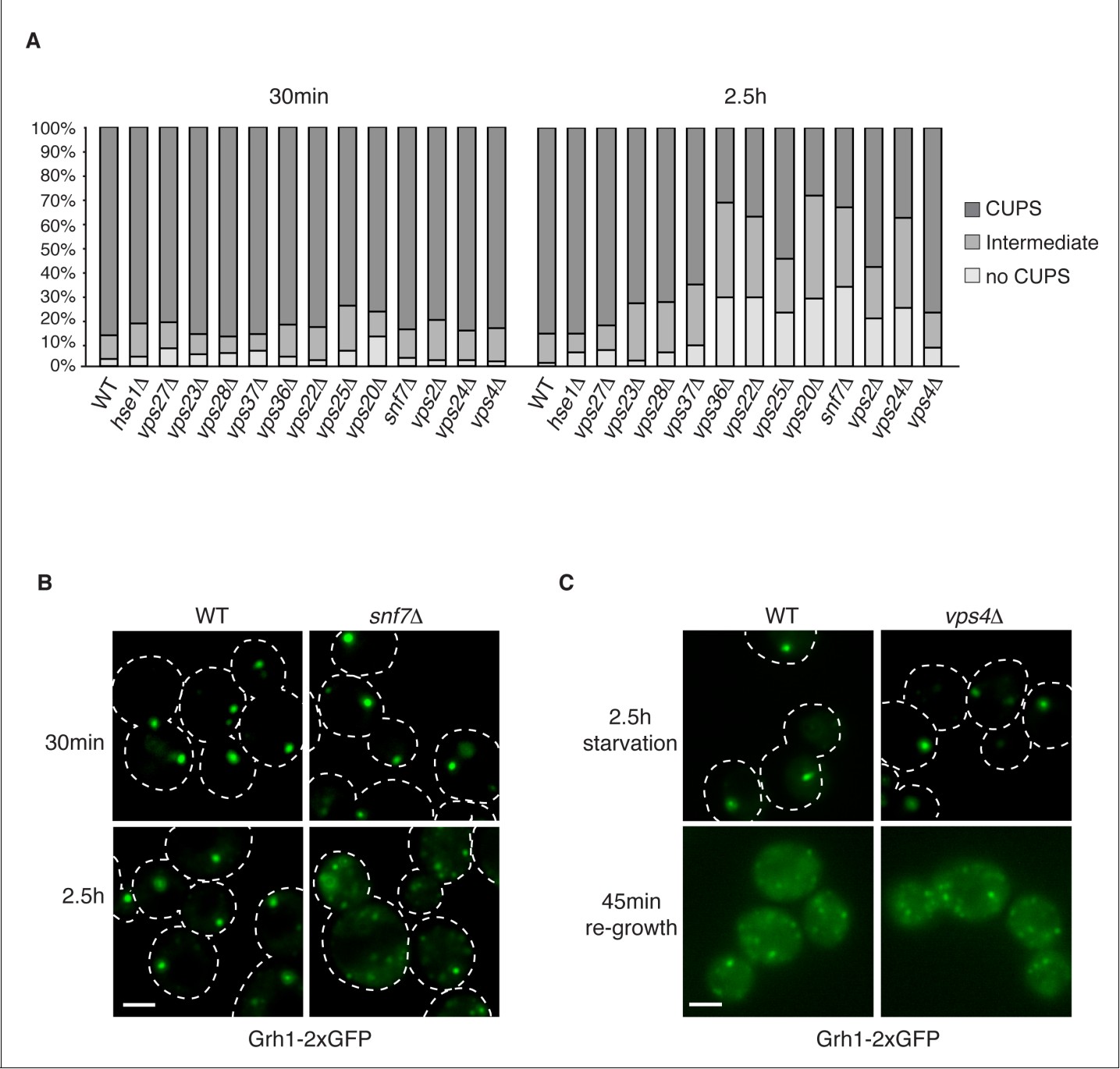

**Figure 3.** The involvement of ESCRTs in CUPS formation, stability and disassembly. (**A**) Wild type and the indicated deletion strains expressing Grh1-2xGFP were grown to mid-logarithmic phase, washed twice, and incubated in 2% potassium acetate for 30 min and 2.5 hr to assess CUPS biogenesis and stability, respectively. Cells were grouped in 3 classes: CUPS (1–3 large punctae per cell), intermediate (large and small punctae), no CUPS (multiple small punctae). Between 50–200 cells were counted for each strain/condition from 3 independent experiments. (**B**) Effect of loss of Snf7 of ESCRT-III at 30 min and 2.5 hr starvation. (**C**) Wild type and *vps4Δ* cells were starved for 2.5 hr as in (**A**) to allow CUPS formation. Cells were washed once and cultured in rich media for 45 min. Scale bars = 2 μm.

GFP expression displayed altered CUPS morphology (data not shown), we therefore focused on cells that exhibited normal CUPS and found that ~15% of such cells co-localized Snf7-GFP to Grh1-2xmCherry containing CUPS (*Figure 4B*). To further ascertain the location and dynamics of Snf7 to Grh1 containing CUPS, we performed time-lapse confocal imaging during starvation. Our results

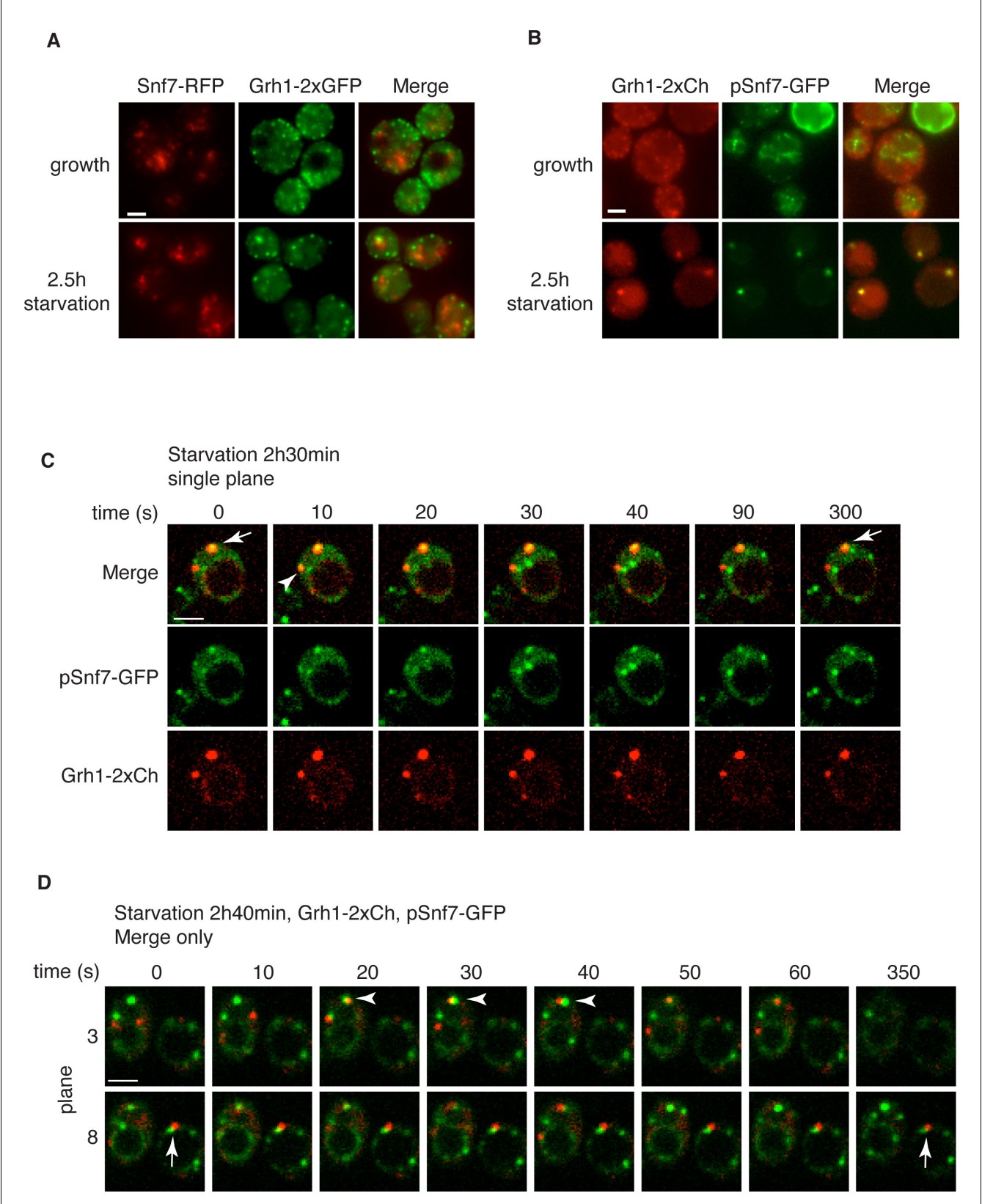

**Figure 4.** Snf7 localizes transiently to CUPS. (**A**) Genomically integrated Snf7-RFP and Grh1-2xGFP were visualized by fluorescence microscopy during growth in mid-logarithmic phase and after incubation in 2% potassium acetate for 2.5 hr. (**B**) Snf7-GFP was expressed exogenously in cells expressing
*Figure 4 continued on next page*

Figure 4 continued

genomically integrated Grh1-2xmCherry (Grh1-2xCh) and visualized by fluorescence microscopy during growth in mid-logarithmic phase and after incubation in 2% potassium acetate for 2.5 hr. (C–D) The same cells from (B) were visualized over time throughout starvation by confocal spinning disk microscopy. Cells with Grh1-2xmCherry on CUPS and Snf7-GFP expression were quantitated for frequency and duration of co-localization and/or overlap of the two structures. Rapid (less than 30 s) and stable (more than 30 s) co-localization examples are indicated by arrowheads and arrows, respectively. All scale bars = 2 μm.

The following figure supplement is available for figure 4:

**Figure supplement 1.** GFP-Cps1 transport is unaffected in pSnf7-RFP expressing cells.

showed that Snf7 and Grh1 exist on separate compartments for the majority of the time during starvation. However, these compartments periodically contacted and/or coalesced (*Figure 4C–D*). Quantitative analysis of cells with Snf7-GFP expression and Grh1-2xmCherry on CUPS indicated that co-localization occurred transiently only after 1 hr of starvation (see also *Figure 6A*). Between 1 and 2.5 hr of starvation, analysis of 62 cells revealed Snf7 and Grh1 compartments co-localized or overlapped rapidly 17 times in 17 different cells for 10–20 s. Interestingly, between 2.5 and 3 hr of starvation, the time when Acb1 secretion peaks, analysis of 96 cells revealed 37 co-localization events in 25 cells, meaning co-localization was observed more than once per cell in some cases. Of these 37 co-localization events, 22 were rapid, lasting only 10–30 s (examples shown in *Figure 4C–D*, arrowheads), while 15 were stable, sometimes lasting for up to 10-minutes (examples shown in *Figure 4C–D*, arrows). In summary, the frequency and duration of co-localization increased throughout starvation, correlating with both the timing of Acb1 secretion and the requirement of Snf7 in CUPS stability but not with the initial steps of CUPS biogenesis (*Figures 2* and *3*). Due to photo bleaching, it was not possible to visualize the same cell throughout the time-course of starvation. Despite this, from the combined data we propose that all CUPS at one point contact or fuse with the Snf7 containing compartment and this is necessary for the final step of maturing CUPS into a secretion competent organelle. In the absence of Snf7, this final maturation might be blocked and Grh1 containing CUPS disintegrate. One possibility is that Snf7 is required for sealing of the membranes that compose CUPS to generate a compartment sealed from the cytoplasm.

## Recruitment of Snf7 to membranes for Acb1 secretion and MVB pathway share a similar mechanism

The ordered assembly of ESCRT complexes for sorting of transmembrane cargoes into MVBs is well defined (reviewed in *Hurley and Emr, 2006*; *Piper and Katzmann, 2007*). ESCRT-0 binds and recognizes ubiquitinated cargoes that in turn recruit ESCRT-I, followed by ESCRT-II (*Katzmann et al., 2003*; *Kostelansky et al., 2006*). Vps20 of ESCRT-III binds directly to Vps25 of ESCRT-II and initiates ESCRT-III filament formation by recruiting Snf7 and nucleating its homo oligomerization (*Teis et al., 2008*; *2010*). Binding of Vps2 and Vps24 caps ESCRT-III filament formation and followed by recruitment of Vps4 to constrict the necks of budding ILVs and to finally disassemble the ESCRT-III filaments (*Babst et al., 1998*; *Saksena et al., 2009*; *Adell et al., 2014*).

Is the attachment of Snf7 to CUPS similar to the order of events as in MVB biogenesis? Snf7-RFP was expressed exogenously from its own promoter in wild type and ESCRT deleted Grh1-2xGFP expressing strains. The heterogeneity in Snf7-RFP expression levels was more pronounced in the deletion strains compared to wild type cells. A small percentage of mutant cells with very high Snf7-RFP levels had aberrant accumulations and were excluded from further investigation. Snf7 recruitment to membranes in growth conditions was as expected for the various ESCRT deletions (*Figure 5*). We found that loss of ESCRT-I (Vps23, Vps28), -II (Vps36, Vps25) or Vps20 (ESCRT-III) proteins abolished the vacuolar membrane localization of Snf7 and resulted in a more diffuse cytosolic pattern with an occasional faint spot (*Figure 5*). Loss of ESCRT-III components Vps2 or Vps24 resulted in Snf7 localization that was very similar to wild type cells, with vacuole membrane accumulation and an occassional discrete spot. Cells lacking Vps4 displayed a distinct Snf7 localization compared to other ESCRT deletions. In this instance, Snf7 accumulated only in punctate structures and not the vacuole membrane or cytosol (*Figure 5*).

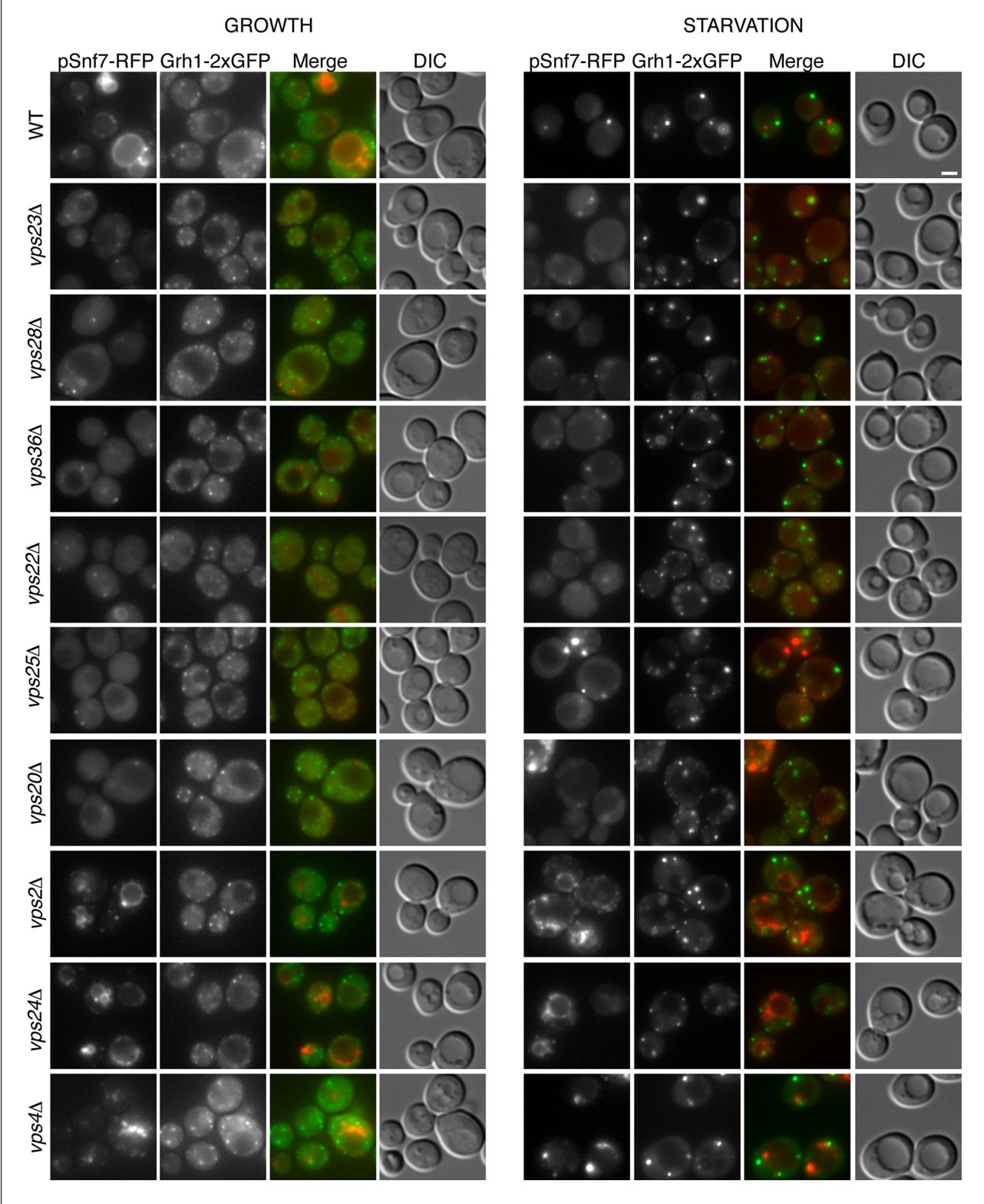

**Figure 5.** Recruitment of Snf7 to membranes for Acb1 secretion and MVB pathway share a similar mechanism. Snf7-RFP was expressed exogenously from its own promoter in wild type and the indicated ESCRT deletion strains expressing Grh1-2xGFP. Cells were visualized by fluorescence microscopy during growth in mid-logarithmic phase (growth) and after incubation in 2% potassium acetate (starvation) for 2–2.5 hr. Scale bar = 2 μm.

We then monitored Snf7 location in ESCRT deleted cells starved for 2–2.5 hr, which is the time with frequent location of Snf7 to Grh1 containing CUPS (*Figure 5*). In wild type cells, Snf7-RFP localization changes to 1–3 prominent foci with reduced vacuole membrane staining (see also *Figure 4B–D*). As mentioned above, we observed mild perturbation in CUPS formation in wild type cells expressing Snf7-RFP. Similarly, Snf7-RFP expression exacerbated the defects in CUPS morphology in most ESCRT mutants (*Figure 5*). Even loss of ESCRT-I components, which normally did not appear to affect CUPS morphology, resulted in more cells with incomplete CUPS formation upon Snf7-RFP expression. Similar to the situation in growth conditions, ESCRT-I mutants displayed more cytosolic distribution of Snf7-RFP and there was no vacuole membrane localization compared to wild type cells. This suggests that, as in growth, Snf7 recruitment to membranes is impaired in these cells upon starvation. However, Snf7 was recruited to 1–2 foci in some cells, although less efficiently, and this could, at times, be co-localized to Grh1 containing CUPS (*Figure 5*). This is expected, as loss of Snf7 recruitment should lead to breakdown of CUPS. The CUPS morphology was severely affected in ESCRT-II mutants and *vps20Δ* cells. Snf7 was mostly cytosolic in these cells, similar to ESCRT-I mutants, however much larger accumulations in 1–2 dots were often observed, but these aberrant structures did not appear to co-localize with Grh1. Such accumulations were also occasionally observed in growth conditions in these deletion strains and we suggest the corresponding structures are dysfunctional that result from sequestration of Snf7 in the absence of the nucleator Vps20. Snf7-RFP in cells lacking Vps24 or Vps2 did not change localization upon starvation and was retained mostly at the vacuole membrane and multiple small foci (*Figure 5*). Notably, Snf7 was not detected in larger foci as in wild type cells (*Figure 5*). In *vps4Δ* cells Snf7-RFP localization did not change dramatically when compared to growth and again displayed a unique localization when compared to other ESCRT deletions. Cells with very high Snf7-RFP levels also altered CUPS morphology, similar to ESCRT-I mutants, however, unlike in *vps2/24Δ* cells, Snf7 localized mostly to larger foci that could be observed to co-localize with Grh1 at times (*Figure 5*).

Altogether the data imply that recruitment of Snf7 to a structure for unconventional secretion follows a similar hierarchy, where ESCRT-I and II enhance Snf7 recruitment and Vps20 acts as the nucleator. Loss of Vps2 or Vps24 resulted in Snf7 immobilization upon starvation which could mean Snf7 was sequestered in growth before starvation or Vps2/Vps24 are directly required for Snf7 relocalization in starvation. Snf7 did localize to punctate elements in *vps4Δ* cells, albeit not exactly as in wild type cells, but as CUPS morphology was not affected and Snf7 co-localization to CUPS could be observed this begins to explain the independence of Vps4 in this process.

## Snf7 recruitment to CUPS and Acb1 secretion are accelerated in *vps4Δ* cells

We wanted to examine in more detail the recruitment of Snf7 to CUPS during starvation in the absence of Vps4. We therefore performed time-lapse confocal imaging as in *Figure 4* and found Snf7 was recruited to CUPS membranes much earlier in starvation than in wild type cells (*Figure 6A*). We examined wild type and *vps4Δ* cells exogenously expressing Snf7-RFP during the first hour of starvation. In wild type cells there was no detectable localization of Snf7-RFP to CUPS, consistent with our findings in *Figure 4*. Specifically, analysis of 65 wild type cells between 10 and 50 min of starvation revealed no co-localization of Grh1 and Snf7 (*Figure 6A* – WT 25 min). In fact, Snf7 foci had not completely formed as later in starvation, while in *vps4Δ* cells Snf7 localized almost exclusively to such foci (*Figure 6A*). Analysis of 72 *vps4Δ* cells between 10 and 50 min of starvation revealed Snf7 localized to CUPS in 20 cells (in 2 cases twice per cell), usually rapidly, for 10–30 s (example shown in *Figure 6A* – *vps4Δ* 15 min – arrowheads). We observed 2 stable co-localization events but these were associated to larger Snf7 accumulations (example shown *Figure 6A* – *vps4Δ* 20 min – arrow). Later in starvation there was no appreciable difference in the frequency or duration of Snf7 localization to CUPS in wild type versus *vps4Δ* cells (data not shown). Therefore Snf7 recruitment to CUPS is accelerated in *vps4Δ* cells.

This prompted us to also examine Acb1 secretion early in starvation. Indeed, we observed a modest acceleration of Acb1 secretion in *vps4Δ* cells. After 1 hr starvation wild type cells had secreted very little Acb1, whereas *vps4Δ* cells secreted more than two fold more Acb1 at this time point (*Figure 6B–C*). We had consistently observed a trend of slight hyper-secretion of Acb1 in *vps4Δ* cells at 2.5 hr starvation (*Figure 2E–F* and *Figure 6B–C*), which could be explained as a result of an increased kinetics of Snf7 recruitment to CUPS and thus an increased rate of Acb1 export.

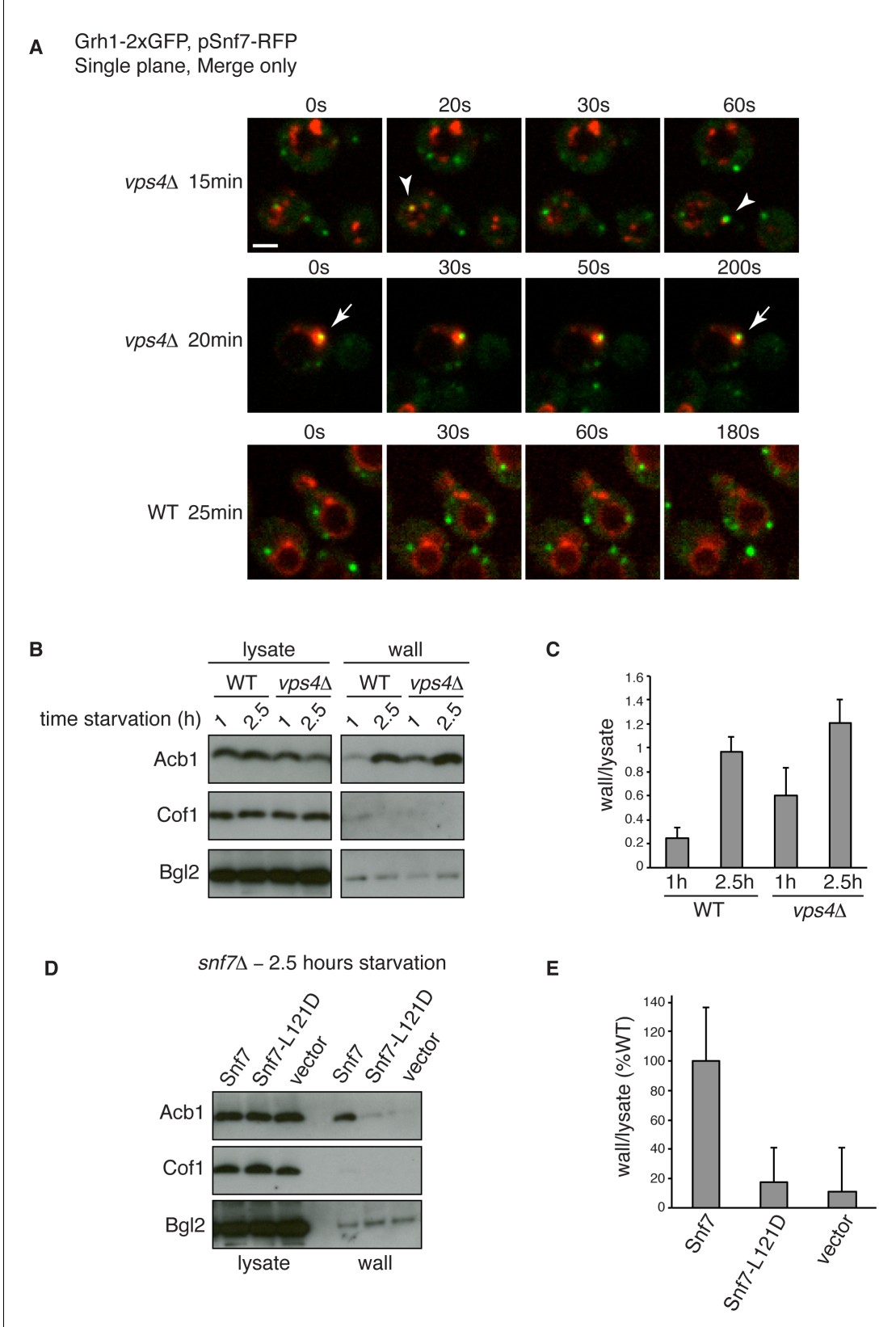

**Figure 6.** Snf7 recruitment to CUPS and Acb1 secretion are accelerated in vps4Δ cells. (**A**) Snf7-RFP was expressed exogenously from its own promoter in wild type and *vps4Δ* cells expressing Grh1-2xGFP. Cells were starved and immediately visualized by time-lapse confocal microscopy (Scale bar = 2

*Figure 6 continued on next page*

Figure 6 continued

µm). (**B–C**) Wild type and *vps4Δ* were grown to mid-logarithmic phase, washed twice, and incubated in 2% potassium acetate for 1 hr and 2.5 hr. Cell wall proteins were extracted as before and analyzed by western blotting. Acb1 levels were quantified and the ratio of wall/lysate Acb1 was determined. n=3 (**D–E**) Snf7 polymerization is required for Acb1 secretion. *snf7Δ* cells expressing wild type Snf7, Snf7-L121D or empty vector were grown to mid-logarithmic phase, washed twice, and incubated in 2% potassium acetate for 2.5 hr. Cell wall proteins were extracted as before and analyzed by western blotting. The ratio of wall/lysate Acb1 was determined (n=3).

To further explore the function of Snf7 for Acb1 secretion we tested a point mutant (L121D) that is unable to homooligomerize and is therefore defective in ESCRT-III filament formation (*Saksena et al., 2009*). We expressed wild type or mutated Snf7 in *snf7Δ* cells and examined Acb1 secretion after 2.5 hr of starvation. Cells expressing empty vector or Snf7-L121D were unable to secrete Acb1, in contrast to cells expressing wild type Snf7 (*Figure 6D–E*). Therefore the ability of Snf7 to polymerize into filaments is required for Acb1 secretion, although Vps4 mediated disassembly is not. In fact, inhibiting the latter leads to slightly accelerated secretion.

## Ultrastructure of CUPS and the involvement of Snf7 in their stability

Immunoelectron microscopy of cells incubated in potassium acetate revealed the presence of Grh1 to a cup-shaped membranous compartment and a collection of small vesicles (*Bruns et al., 2011*). In order to better characterize- in 3D - the Grh1 containing membranes, we made use of the correlative light electron microscopy procedure (CLEM). Cells expressing Grh1-2xmCherry were subjected to high pressure freezing and thin sectioning. The 300 nm sections were first visualized by fluorescence microscopy to locate areas of interest to be analyzed by electron tomography microscopy, followed by three-dimensional reconstruction of the Grh1 containing membranes. In growing cells, as expected, Grh1 was found predominantly in the vicinity of typical Golgi-like cisternae and vesicles that were indistinguishable in wild type and *snf7Δ* cells (*Figure 7A* – WT growth, *Figure 7—figure supplement 1* and data not shown).

We identify CUPS as a convoluted network of tubules and vesicles with an overall spheroidal shape, a cavity at the center, and an average diameter of ∼200 nm (14 of 16 tomograms at 2.5 hr starvation) (*Figure 7A* – WT starvation 1 and *Video 1*). We followed the development of CUPS, based on the location of Grh1-2xmCherry at 45 min and 2.5 hr of starvation. At 45 min of starvation the Grh1 containing structures were more heterogeneous in size, but in general became larger throughout starvation (*Figure 7A–B* and *Figure 7—figure supplement 2*). As the structure became larger, the 'sphere' appeared smoother, or less fenestrated. Of particular note was the presence of a large fenestrated cisternae or saccule that seemed to engulf the Grh1-positive membranes (*Figure 7A* – WT starvation 2 and *Video 2*). This was present in just 15% of CUPS at 45 min starvation and was increased to 50% of the CUPS structures at 2.5 hr (*Figure 7B*). The 'engulfed' CUPS were on average ∼200 nm in size.

In cells lacking Snf7 at 45 min of starvation there were no significant differences found in the organization of Grh1-positive membranes, as we observed by fluorescence microscopy (*Figure 7B* and *Figure 7—figure supplement 2*, see also *Figure 3A–B*). However, after 2.5 hr of starvation more than 80% of the Grh1-positive membranes were identified as small vesicles (*Figure 7A* – *snf7Δ* starvation). Moreover, the large saccule, which was observed associated with Grh1 containing membranes in wild type cells, was never identified in *snf7Δ* cells at 2.5 hr starvation (*Figure 7A–B*).

## Acb1 is contained in CUPS

Our data shows that approximately 1% (roughly 1400 molcules of Acb1 per cell) are secreted upon nutrient starvation (*Figure 1*). Thin sections of fixed yeast cultured in potassium acetate for 2.5 hr were visualized by immunoelectron microscopy with anti-GFP and anti-Acb1 specific antibodies that can bind and detect the cognate protein only when the latter is concentrated. The results reveal three major features. 1, Grh1-2xGFP (15 nm gold) is contained in tubules and vesicles, 2, Grh1 is proximal to a cup-shaped membrane, and 3, Grh1 is encased in a saccule, which contains Acb1 (10 nm gold) (*Figure 8*).

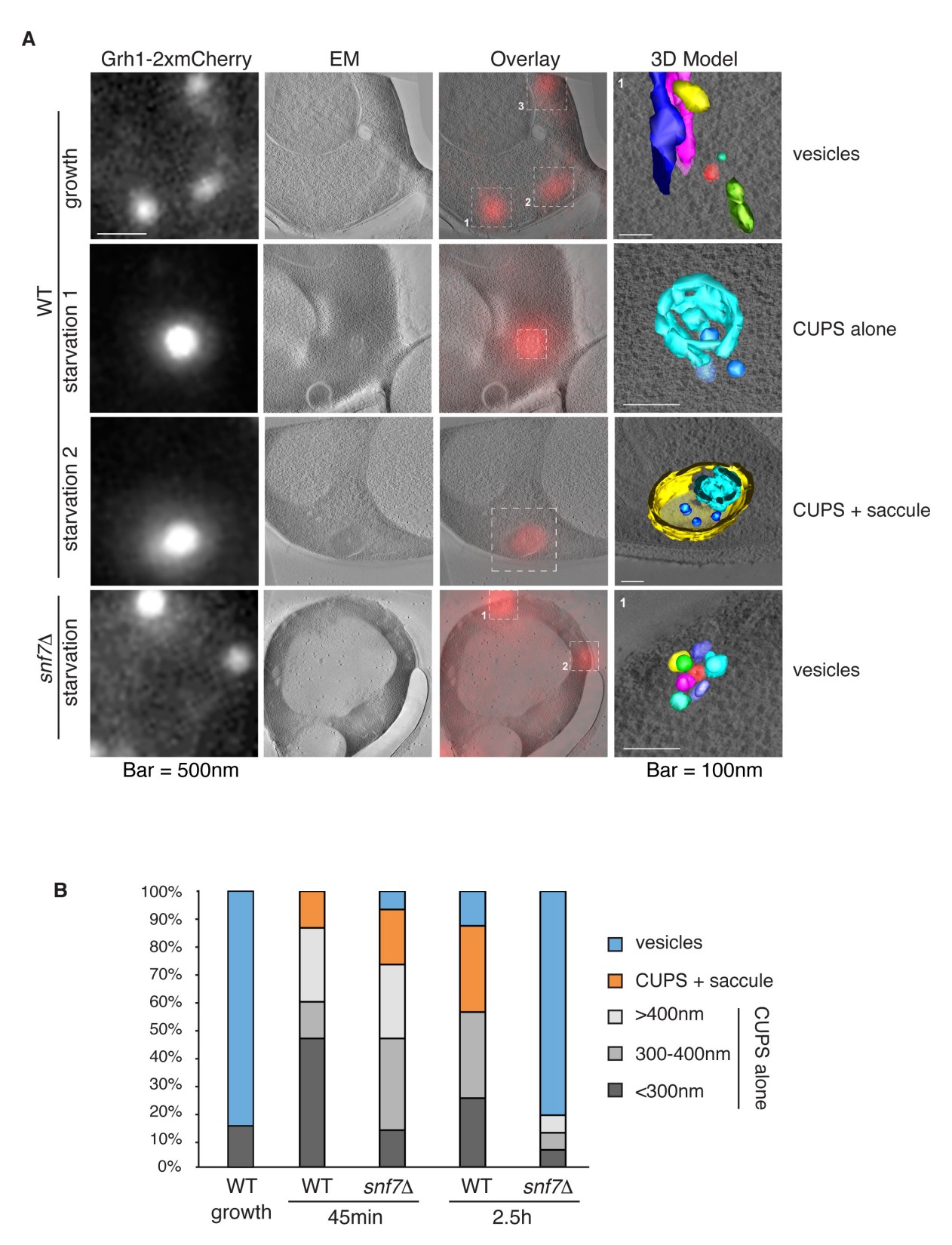

**Figure 7.** Ultrastructure of CUPS and the involvement of Snf7 in their stability. CUPS are revealed as a spheroidal collection of highly curved membranes of average 200 nm diameter (WT starvation 1, 3D Model). (**A**) Grh1-2xmCherry expressing cells were grown to mid-logarithmic phase,

*Figure 7 continued*

washed twice, and incubated in 2% potassium acetate for 2.5–3 hr. Growing or starved cells were subjected to cryofixation and correlative light and electron microscopy (CLEM) (see Materials and methods). High magnification tomograms were acquired and 3D models were reconstructed of the membranes positive for mCherry signal. In the case of wild type cells, 13 tomograms were acquired in growth and 16 during starvation. For *snf7Δ* cells, 14 tomograms were acquired in starvation (the organization of Grh1-2xmCherry positive membranes during growth was indistinguishable from wild type). In 50% of CUPS structures identified in wild type cells a saccule surrounding CUPS was observed (WT starvation). (B) Classification of Grh1-positive membranes and measurements of CUPS structures at 45 min and 2.5 hr starvation. At 45 min starvation 12 tomograms each from wild type and *snf7D* cells were analyzed. 'Vesicles' refers to vesicles and small cisternae. The diameter of the CUPS structures was measured along the longest axis. In the case of 'CUPS + saccule' the CUPS structures were small, with an average diameter of 200 nm.

The following figure supplements are available for figure 7:

**Figure supplement 1.** Remaining 3D models of Grh1-positive membranes from *Figure 7A*.

**Figure supplement 2.** Examples of CLEM analysis from wild type and *snf7Δ* cells at 45 min of starvation.

Based on this data alone, it would be inaccurate to propose these as sequential steps in the pathway of Acb1 secretion, but these images, combined with the CLEM analysis are suggestive of a possible pathway for the formation of CUPS. The initial Grh1 positive tubulo-vesicular clusters resemble the typical mammalian ERGIC compartment (*Figure 8* – Stage 1, *Figure 9* – immature CUPS). At this stage Acb1 is rarely found in these structures. In the next stage, a tubular structure (probably a sheet like structure in 3D) approaches and initiates the engulfment of the Grh1 positive immature CUPS (*Figure 8* – Stage 2, white arrows). We believe this to be the saccule identified by our CLEM analysis (*Figure 7A* – WT starvation 2 and *Video 2*). This Grh1 encasing compartment also contains Acb1 (*Figure 8* – Stage 3, *Figure 9* – stable CUPS).

## Discussion

The development of a new quantitative assay to measure secretion of Acb1, a protein of 87 amino acids, is a significant step forward. Now it is possible to directly identify proteins required for Acb1 secretion and to distinguish them from those with a role in post-translational modifications to produce a functional bioactive peptide of 34 amino acids in the extracellular space. This assay has also allowed us to rule out the possibility that the effect of various genes on the release of Acb1 was indirect due to cell wall stability. The assay has revealed that ~1% of total full length Acb1 is secreted in response to a starvation-induced signal (*Figure 1*).

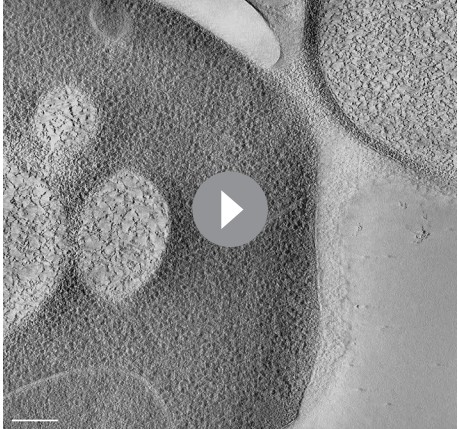

**Video 1.** Morphology of CUPS Full tomogram and 3D reconstruction of Grh1-2xmCherry positive membranes corresponding to *Figure 7A* – WT – starvation 1 – 'CUPS alone'.

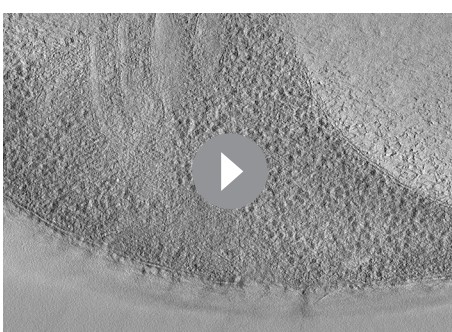

**Video 2.** Morphology of stable CUPS Full tomogram and 3D reconstruction of Grh1-2xmCherry positive membranes corresponding to *Figure 7A* – WT – starvation 2 – 'CUPS + saccule'.

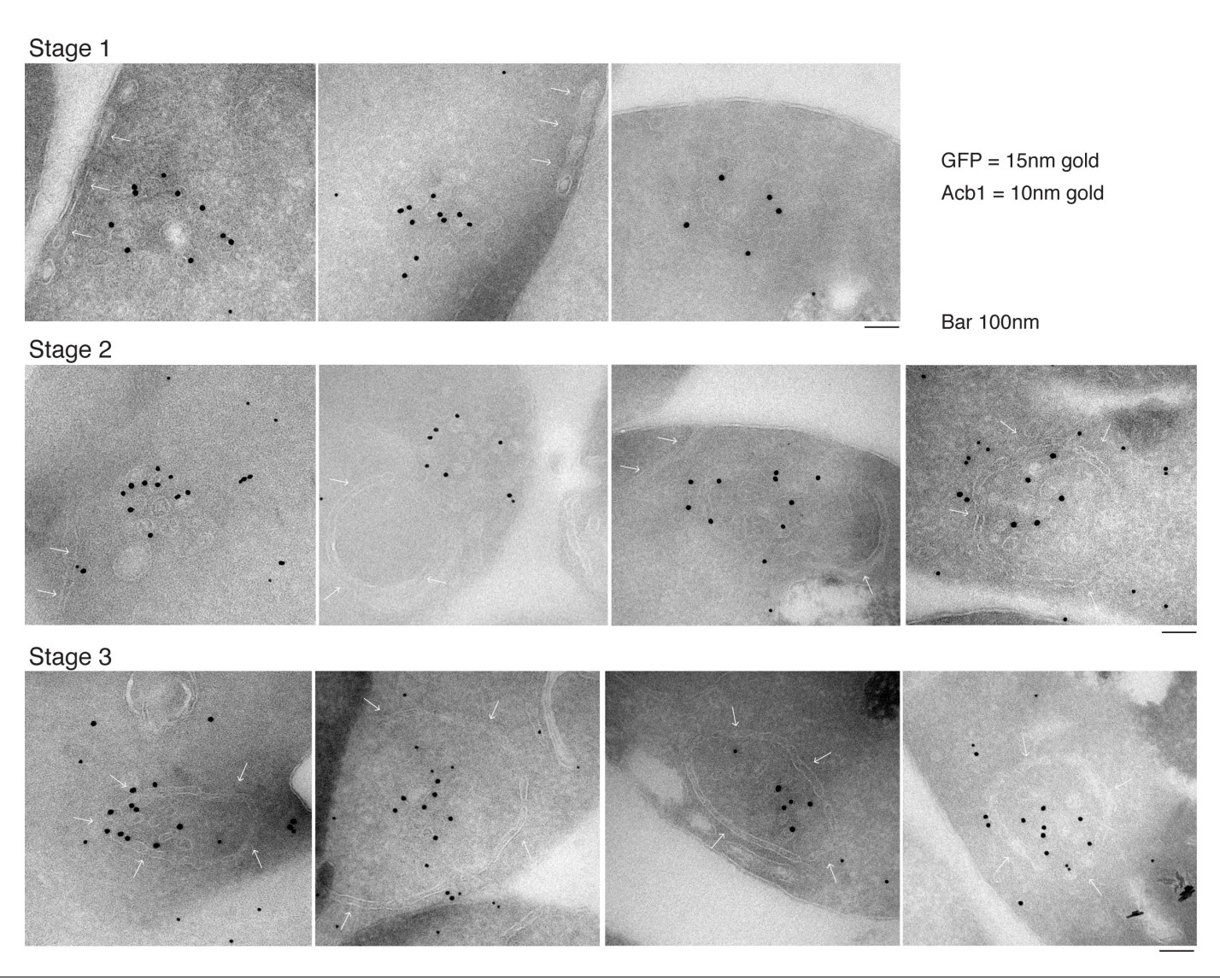

**Figure 8.** Immunoelectron microscopy identifies Acb1 in stable CUPS. Grh1-2xGFP expressing cells were starved for 2.5 hr and processed for immunogold labeling and electron microscopy (see Materials and methods). GFP = 15 nm gold, Acb1 = 10 nm gold. Stage 1 – Grh1 labels tubulo-vesicular structure. Stage 2 – A sheet/saccule approaches and initiates engulfment of the Grh1 labelled structure (white arrows). Stage 3 – The Grh1 structure is completely engulfed and contains Acb1 – mature or stable CUPS.

Our results reconfirm the conserved role of the GRASP ortholog, Grh1, in the general process of unconventional protein secretion (*Figure 2A–B*) (*Kinseth et al., 2007*; *Zhang et al., 2015*; *Duran et al., 2010*; *Manjithaya et al., 2010*; *Dupont et al., 2011*; *Gee et al., 2011*; *Schotman et al., 2008*).

## The role of ESCRTS in unconventional secretion

There are numerous reports of exosomes as vehicles for the release of cytoplasmic proteins in the extracellular space. These extracellular vesicles are derived from the fusion of multivesicular bodies (MVBs) to the cell surface (reviewed in *Bobrie et al., 2011*). While a number of issues on the role of exosomes in secretion remain unclear, our new findings strongly indicate that canonical MVBs are not involved in Acb1 secretion. Acb1 secretion requires ESCRT-I, -II and -III proteins but not ESCRT-0 or Vps4 proteins (*Figure 3*). The lack of ESCRT-0 involvement is not surprising as its main function is to recognize transmembrane MVB cargoes. Moreover, other ESCRT dependent membrane fission

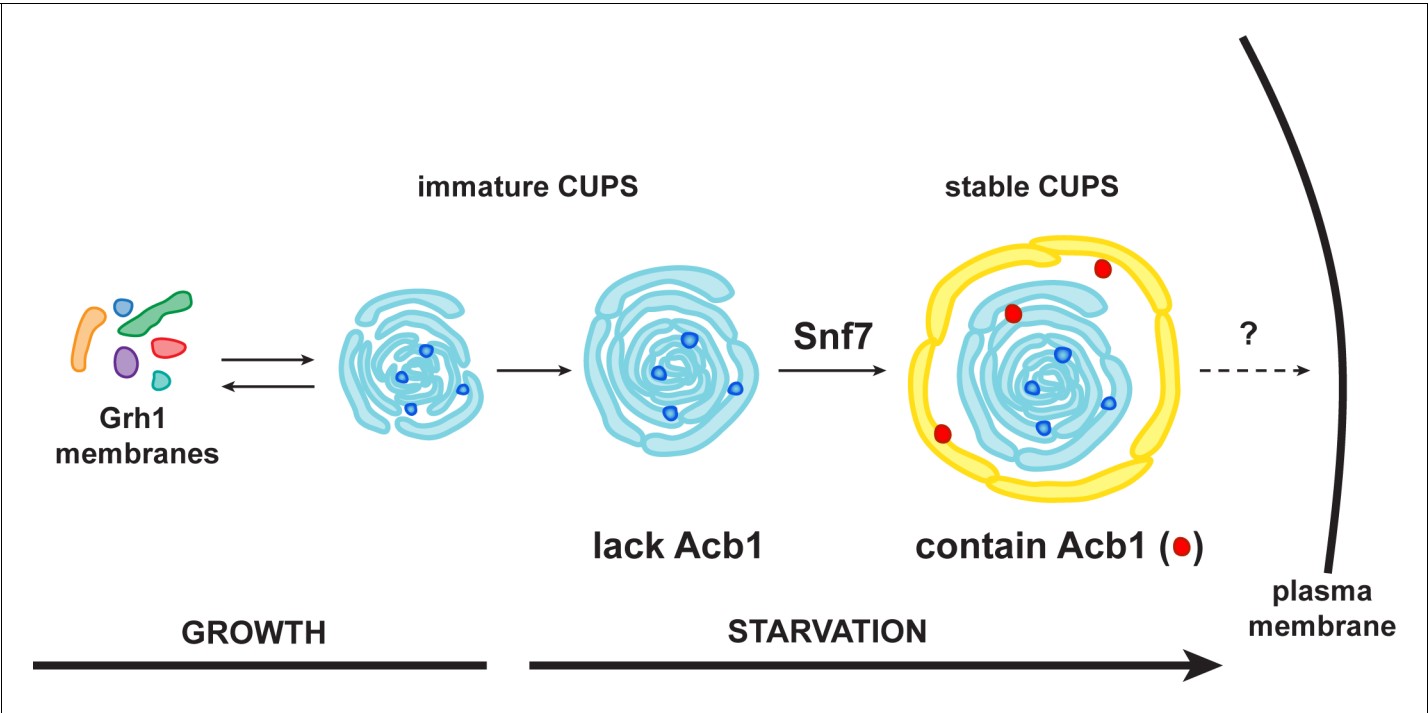

**Figure 9.** A schematic presentation of steps in Acb1 secretion. Grh1 containing vesicles and tubules assemble into foci that are clearly visible as one to two spots per cell. This collection of membranes are formed and consumed constitutively during growth (immature CUPS). However, upon shifting cells to starvation medium, the immature CUPS become encased in a saccule (yellow membrane) by an Snf7 mediated reaction. We call this membrane-bounded compartment 'stable CUPS'. This stable form of CUPS is in fact found to contain Acb1. Stable CUPS are quite different from a standard MVB. The stable CUPS are double membrane bounded and contain internal membranes of different sizes and shapes. An MVB on the other hand, is composed of a single bilayer containing uniformed sized vesicles. The key difference also is that Vps4 is required for the formation of an MVB, but not for the Snf7 dependent engulfment of Grh1 containing immature CUPS. The stable CUPS then release Acb1 to the exterior of the cells.

events, such as HIV budding, cytokinesis and nuclear envelope re-sealing after mitosis also require only a subset of ESCRT proteins and are independent of ESCRT-0 (*Morita et al., 2007*; *Garrus et al., 2001*; *Vietri et al., 2015*). However, the function of Vps4 is essential for all aforementioned ESCRT-dependent functions. The pathway of Acb1 secretion requires ESCRT-III, but is independent of Vps4.

## ESCRT-III/Snf7 functions in the production of stable and functional CUPS

Grh1 relocates from ER exit site/intermediate compartment like structures in growing yeast cells to a new compartment called CUPS upon starvation (*Bruns et al., 2011*; *Cruz-Garcia et al., 2014*). The lack of requirement for ESCRT-0 and Vps4 in CUPS formation correlates with our secretion data indicating these proteins, and thereby the MVB pathway, are not required for Acb1 secretion. Our findings also show that loss of Vps4 promotes an earlier location of Snf7 to CUPS and this correlates with faster Acb1 secretion (*Figure 6*).

Deletion of Snf7 did not affect the initial assembly of CUPS from small Grh1 containing vesicles, but these immature CUPS were unstable, did not proceed to become fully mature CUPS and finally broke into small clusters of vesicles (*Figures 3* and *7*). This indicates a role of Snf7/ESCRT-III in maintaining the structural integrity of CUPS. We observed that Grh1 and Snf7 reside on different compartments that contact and at times coalesce (*Figures 4* and *6*). Based on our findings, we suggest that Grh1 containing membranes contact and are encased in Snf7 containing compartments and this produces stable CUPS. Based on the CLEM data we suggest that Grh1 containing immature CUPS likely form constitutively, even during growth, as evident by the presence of CUPS like structures in 2 of 16 tomograms during growth, but are rapidly dismantled if their function is not required for the purpose of unconventional secretion (*Figure 7*). Upon starvation their consumption to the ER is

blocked and they mature - by ultimately being encased in a membrane by Snf7 dependent reaction to form stable CUPS (*Figures 7–9*).

## Stable CUPS contain Acb1

We have been unable to detect Acb1 in the immature Grh1 containing vesicles and tubules. However, once they are encased in a saccule by a process that correlates with co-localization of Grh1 and Snf7, the ensuing compartment is found to contain Acb1 (*Figure 8*). We do not know the origin of the saccular membrane, but our findings strongly suggest that acquisition of Acb1 into the secretory pathway is post-production of Grh1 containing vesicular membranes. One possibility is that Acb1 is attached to the saccule that engulfs Grh1 containing membranes and this casing of saccule is sealed by an Snf7 dependent reaction to produce a compartment whose contents are separated from the cytoplasm. This doubled membrane bounded multivesicular / multilamellar compartment is clearly morphologically distinct from an MVB, that is composed of a single membranes containing uniformed sized vesicles. Moreover its formation is different from the mechanism by which cells produce an MVB. The role of Vps4 in the biogenesis of these compartments is the most significant difference. While Vps4 is required for MVB biogenesis, it is not involved in mature CUPS formation. We call this compartment stable CUPS to distinguish them from the collection of Grh1 containing vesicles and tubules that are observed at 30 min post starvation (*Figure 9*). The stable CUPS containing sealed Acb1 are then the source for the release of Acb1 into the exterior of the cell.

## Similarities of requirements for Acb1 secretion and cells response to high pH

Cells exposed to alkaline pH activate the Rim signaling pathway, which also depends on a subset of ESCRT proteins. Briefly, the arrestin-like protein Rim8 is recruited to plasma membrane via the plasma membrane sensor Rim21 (*Obara et al., 2012*; *Obara and Kihara, 2014*; *Herrador et al., 2010*). Rim8 directly recruits Vps23 of ESCRT-I, bypassing ESCRT-0 (*Herrador et al., 2010*). This ultimately leads to ESCRT-III dependent recruitment of a protease (Rim13) that cleaves a transcription factor (Rim 101), which then regulates the genes that provide protection from the elevated pH (*Lamb et al., 2001*; *Weiss et al., 2009*). This process is independent of ESCRT-0 but always active in the absence of Vps2, Vps24 or Vps4, which cause constitutive recruitment and cleavage of Rim101 (*Hayashi et al., 2005*). Acb1 secretion and CUPS formation is also independent of ESCRT-0, but do require Vps2/24 proteins. Although Vps4 is not required for Acb1 secretion or CUPS formation, the loss of Vps4 does not induce Acb1 secretion or CUPS formation in growing conditions. We did, however, observe modest acceleration of Acb1 secretion and Snf7 recruitment to CUPS upon starvation in *vps4Δ* cells. It is therefore unclear whether the conditions that trigger Acb1 secretion are similar to cellular response to pH alterations. Moreover, we have tested loss of Rim8, Rim21 and Rim101 directly for Acb1 secretion and unfortunately these gene deletions caused extensive cell lysis thereby precluding their evaluation in Acb1 secretion by our cell wall extraction assay (data not shown). Clearly, the mechanism by which cells sense and signal in response to nutrient starvation to generate CUPS and promote Acb1 secretion is an interesting challenge and might well share some genetic requirements of the other well known signaling pathways such as the high pH triggered Rim signaling pathway.

## Secretion of Acb1 and IL-1ß similar with differences

It has recently been shown that siRNA dependent knockdown of GRASP55, GRAPS65, Hrs (Vps27, ESCRT-0) and Tsg101 (Vps23, ESCRT-I) affected unconventional IL-1ß secretion (*Zhang et al., 2015*). This confirms some of the data previously reported for Acb1 secretion (*Duran et al., 2010*; *Manjithaya et al., 2010*). Based on their data, Schekman and colleagues have suggested that IL-1ß is translocated into a vesicle, which grows into a phagophore and is converted into an autophagosome that contains IL1ß between the inner and the outer membrane. The autophagosome then either fuses directly with the cell surface or first with an endosome/MVB to generate an amphisome that later fuses with the plasma membrane to release soluble IL-1ß (*Zhang et al., 2015*). Although, many mechanistic questions remained unsolved: the authors have not reported the requirement of other ESCRTS, the genes required for complete formation of an autophagosome, the proposed translocon for transferring IL-1ß across the membrane, the involvement of endosomes, amphisomes,

or the fusion of membranes with the cell surface. Regardless, how does this pathway relate to the secretion of Acb1 in yeast?

Our findings indicate that Grh1 containing membranes, after their production by COPI and COPII independent reaction (*Cruz-Garcia et al., 2014*) are first collected into a focus and then appear more tubulo vesicular (*Figure 9* – immature CUPS). These membranes then grow and appear to be more connected. The phagophores implicated by us previously (*Duran et al., 2010*; *Bruns et al., 2011*), and described recently by Schekman and colleagues, could represent these immature CUPS structures. Our data reveal that Grh1 containing membranes (immature CUPS) are encased in a saccule by Snf7 dependent process. This step is required for full maturation of CUPS. We call this compartment stable CUPS that contain Acb1 (*Figure 9*). We speculate that this compartment might be the functional ortholog of the amphisomes proposed by Schekman and colleagues for IL-1ß secretion. Built into this common theme are also other features with respect to specific cargoes that might utilize different chaperones and means for their capture into the starting stage (the vesicles), which could explain the utility of ESCRT-0 for IL1ß, but not Acb1 secretion. It is important to note that the compartments of this pathway do not need to be identical in organization and shape: the yeast Golgi membranes do not appear anything like the Golgi stacks of higher eukaryotes. So the overall - evolving - pathway of unconventional protein secretion is likely more similar than presently appreciated. Importantly, this pathway does not use the conventional Vps4 and MVB mediated release of exosomes into the extracellular space.

## Materials and methods

### Media, yeast strains and plasmids

Yeast cells were grown in synthetic complete (SC) media (0.67% yeast nitrogen base without amino acids, 2% glucose supplemented with amino acid drop-out mix (SIGMA-Aldrich, St. Louis, MO, USA). All strains are derived from the BY4741 background (*MATa his3Δ1 leu2Δ0 met15Δ0 ura3Δ0*). Deletion strains were from the EUROSCARF collection with individual genes replaced by KanMx4. Strains expressing C-terminally 2xyeGFP- and/or 2xyomCherry-tagged Grh1 were constructed by a PCR-based targeted homologous recombination and have been described previously (*Cruz-Garcia et al., 2014*). The Snf7-RFP strain was a kind gift from Dr. Erin O'Shea (Harvard University). All subsequent strains were generated by mating and sporulation, followed by selection of clones with appropriate markers, and confirmation of haploidy. The pRS415-GFP-Cps1 under the control of its own was provided by David Teis. The pRS416-Snf7-GFP expressed from its own promoter was kindly provided by Dr. Scott Emr (Cornell University, Ithaca, USA). The Snf7-RFP expression construct was generated by PCR amplification of Snf7, with its own promoter, mRFP tag and ADH1 terminator sequences from genomic DNA derived from the Snf7-RFP strain and ultimately cloned into pRS416. Similarly, to generate Snf7-L121D construct, Snf7 with its own promoter and terminator was first amplified from wild type genomic DNA and cloned into pRS416. The latter was mutated using the Gibson assembly method to change codon 121 from CTT to GAT (*Gibson et al., 2009*). SnapGene software (from GSL Biotech, Chicago, IL; available at www.snapgene.com) was used for molecular cloning design.

### Unconventional secretion assay

Yeast cells were grown to mid-logarithmic phase by at a density of 0.003–0.006 $OD_{600}$ in SC medium at 25°C. The following day, when cells had reached $OD_{600}$ of 0.4–0.7 equal numbers of cells (15 $OD_{600}$ units) were harvested, washed twice in sterile water, resuspended in 1.5 mL of 2% potassium acetate and incubated for 2 hr. Concomitant to this, growing cells were diluted in SC medium, continued growing in logarithmic phase and 15 $OD_{600}$ units were harvested as before. The cell wall extraction buffer (100 mM Tris-HCl, pH 9.4, 2% sorbitol) was always prepared fresh before use and kept on ice. To ensure no loss of cells and to avoid cell contamination in the extracted buffer, 2 mL tubes were siliconized with Sigmacote (SIGMA-Aldrich) prior to collection. Cells were harvested by centrifugation at 3000xg for 3 min at 4°C, medium or potassium acetate was removed and 1.5 mL of cold extraction buffer was added. Cells were resuspended gently by inversion and incubated on ice for 10 min, after which they were centrifuged as before, 3000xg for 3 min at 4°C, and 1.3 mL of extraction buffer was removed to ensure no cell contamination. The remaining buffer was removed

and the cells were resuspended in 0.75 mL of cold TE buffer (Tris-HCl, pH 7.5, EDTA) with protease inhibitors (aprotinin, pepstatin, leupeptin [SIGMA-Aldrich]) and 10 µL was boiled directly in 90 µL of 2x sample buffer (lysate). To the extracted protein fraction, 30 µg of BSA (bovine serum albumin [SIGMA-Aldrich]) carrier protein was added and 0.2 mL of 100% Trichloroacetic acid (SIGMA-Aldrich). Proteins were precipitated on ice for 1 hr, centrifuged 16,000xg for 30 min and boiled in 45 µL 2x sample buffer. For detection, proteins (10 µL each of lysate or wall fractions) were separated in a 16.5% Tris-tricine peptide gel (Bio-Rad) allowing separation of Cof1 (15.kDa) from Acb1 (10.kDa), before transfer to nitrocellulose. Rabbit anti-Cof1 antibody was a generous gift from Dr. John Cooper (Washington University in St. Louis, USA). Rabbit anti-Bgl2 was a gift from Dr. Randy Schekman (UC Berkeley, CA, USA). Rabbit anti-Acb1 antibody was generated by inoculating rabbits with recombinant, untagged Acb1, purified from bacteria. Specificity of the serum was confirmed by testing lysates prepared from $acb1\Delta$ cells.

## Epifluorescence microscopy

After incubation in the appropriate medium cells were harvested by centrifugation at 3000 g for 3 min, resuspended in a small volume of the corresponding medium, spotted on a microscope slide, and imaged live with a DMI6000 B microscope (Leica microsystems, Wetzlar, Germany) equipped with a DFC 360FX camera (Leica microsystems) using an HCX Plan Apochromat 100x 1.4 NA objective. Images were acquired using LAS AF software (Leica microsystems) and processing was performed with ImageJ 1.47n software.

## Confocal fluorescence microscopy

After incubation in starvation medium for 20 min, ~0.05 OD600 nm of cells were plated in starvation medium on Concanavalin A–coated (SIGMA-Aldrich) Lab-Tek chambers (Thermo Fisher Scientific, Waltham, MA, USA) and were allowed to settle for 20 min at 25°C. Due to issues of bleaching, fields of cells were continuously imaged up to 10 min throughout starvation. Whole cell Z stacks with a step size of 0.3 µm were continuously acquired (10 s frames) using a spinning-disk confocal microscope (Revolution XD; Andor Technology, Belfast, United Kingdom) with a Plan Apochromat 100× 1.45 NA objective lens equipped with a dual-mode electron-modifying charge-coupled device camera (iXon 897 E; Andor Technology) and controlled by the iQ Live Cell Imaging software (Andor Technology). Processing was performed with ImageJ 1.47n software.

## Correlative light electron microscopy (CLEM)

For CLEM analysis, yeast cells were filtered into a paste and cryoimmobilised with HPM 010 high pressure freezing machine (Bal-Tec, Los Angeles, CA, USA). Freeze-substitution with 0.1% Uranyl Acetate in acetone and embedding in Lowicryl resin was performed in an AFS2 machine (Leica microsystems) as described in (Kukulski et al., 2011). 300 nm thick sections were cut from the polymerized resin block and picked up on carbon coated mesh grids. 50 nm TetraSpeck fluorescent microspheres (Lifetechnologies, Carlsbad, CA, USA) were adsorbed to the grid for the subsequent fiducial-based correlation between light and electron microscopy images. The fluorescence microscopy (FM) imaging of the sections was carried out as previously described (Kukulski et al., 2011; Avinoam et al., 2015) using a widefield fluorescence microscope (Olympus IX81) equipped with an Olympus PlanApo 100X 1.45 NA oil immersion objective. Images were collected with mcherry-specific settings as well as in the green channel (Tetraspecks could be distinguished from the mcherry-specific signal by their fluorescence in both channels). Grids were then incubated with 10 nm protein A-coupled gold beads and stained with Uranyl Acetate and Reynolds lead citrate. Tilt series were then acquired with a FEI Tecnai F30 electron microscope. Lower magnification series (3900X) for tetraspecs-based correlation as well as high magnification series (20000X) were acquired. Tomograms were then reconstructed using the IMOD software package (Kremer et al., 1996). Areas of interest were identified with a fiducial-based correlation performed as described previously using in-house written MATLAB scripts (Kukulski et al., 2011; 2012). Briefly, the position of Tetraspeck microspheres was manually assigned in both the FM and low magnification EM images. The coordinate of fiducials pairs in the two imaging modalities were used to calculate a linear transformation, which allowed to map the coordinates of the fluorescent spot of interest on the electron tomogram. The gold beads were then used to calculate the transformation between low and high magnification

tomograms, therefore allowing the overlay of the FM image with the high resolution tomogram. Once the Grh1-2xmCherry fluorescent signal was mapped on the high magnification tomogram, the membranes in the area of interest were manually segmented with IMOD and a 3D model was reconstructed.

### Cryo immunogold labelling

Samples were processed essentially as described earlier (*Peters et al., 2006*). In brief, yeast cells expressing Grh1-2xGFP cultured in starvation conditions (see above) were fixed with 4% paraformaldehyde in PBS for 2 hr. The cells were then washed with PBS/0.02 M glycine, and resuspended in 12% gelatin in PBS, and then embedded in the same solution. The embedded cells were cut in 1 mm blocks and infiltrated with 2.3 M sucrose at 4°C, mounted on aluminum pins, and frozen in liquid nitrogen. The samples were then sectioned and the ultrathin cryosections were picked up in a mixture of 50% sucrose and 50% methylcellulose and incubated with antibodies to Acb1 and GFP (to monitor Grh1) followed by protein A gold (15 nm and 10 nm respectively) in this sequential order. The labelled sections were then imaged in FEI Tecnai-12 electron microscope.

## Acknowledgements

We would like to thank members of the Malhotra lab, in particular David Cruz-Garcia and Juan Duran, as well as Scott Emr of Cornell University and Chris Burd of Yale for helpful discussions. Margherita Scarpa is thanked for generating the recombinant Acb1 protein used for antibody generation. All confocal imaging was performed at the Centre for Genomic Regulation Advanced Light Microscopy Unit. We acknowledge support from the Spanish Ministry of Economy and Competitiveness, 'Centro de Excelencia Severo Ochoa 2013-2017', SEV-2012-0208. V. Malhotra is an Institució Catalana de Recerca i Estudis Avançats (ICREA) professor at the Center for Genomic Regulation and the work in his laboratory is funded by grants from Plan Nacional (BFU2008-00414), Consolider (CSD2009-00016), and European Research Council (268692). Work was in the Teis lab funded by HFSP-CDA-00001/2010-C, Austrian Science Fund (FWF), Y444-B12, MCBO (W1101-B18), SFB021 (F21). The project has received research funding from the European Union. This paper reflects only the author's views. The Union is not liable for any use that may be made of the information contained therein.

## Additional information

### Competing interests

VM: Senior editor, *eLife*. The other authors declare that no competing interests exist.

### Funding

| Funder | Grant reference number | Author |
| --- | --- | --- |
| European Research Council | 268692 | Vivek Malhotra |
| Plan Nacional | BFU2008-00414 | Vivek Malhotra |
| Consolider | CSD2009-00016 | Vivek Malhotra |
| Human Frontier Science Program | HFSP-CDA-0000½010-C | David Teis |
| Austrian Science Fund | Y444-B12 | David Teis |
| Austrian Science Fund | MCBO (W1101-B18) | David Teis |
| Austrian Science Fund | SFB021 (F21) | David Teis |

The funders had no role in study design, data collection and interpretation, or the decision to submit the work for publication.

### Author contributions

AJC, Contributed to the conception of the project, Writing the manuscript, Developed the secretion assay, Performed most experiments, Analysis and interpretation of data, Contributed unpublished

essential data or reagents; NB, Generated strains and reagents, Performed some experiments, Analysis and interpretation of data, Drafting or revising the article, Contributed unpublished essential data or reagents; MAYA, Acquisition of data, Analysis and interpretation of data, Contributed unpublished essential data or reagents; DT, Analysis and interpretation of data, Drafting or revising the article, Contributed unpublished essential data or reagents; GT, Performed the immunoelectron microscopy, Analysis and interpretation of data; SP, Performed the immunoelectron microscopy, Analysis and interpretation of data, Drafting or revising the article; PR, Performed the CLEM analysis, Analysis and interpretation of data, Drafting or revising the article, Contributed unpublished essential data or reagents; VM, Contributed to the conception of the project, Writing the manuscript, Analysis and interpretation of data

### Author ORCIDs

Amy J Curwin, http://orcid.org/0000-0003-1086-2483
Nathalie Brouwers, http://orcid.org/0000-0002-9808-9394
David Teis, http://orcid.org/0000-0002-8181-0253
Vivek Malhotra, http://orcid.org/0000-0001-6198-7943

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
