## [Decision Letter]

Thank you for submitting your article "ESCRT-III drives the final stages of CUPS maturation for unconventional Acb1 secretion independent of Vps4" for consideration by *eLife*. Your article has been reviewed by three peer reviewers, one of whom, Greg Odorizzi has agreed to share his identity, and the evaluation has been overseen by Randy Schekman as the Senior Editor and Reviewing Editor.

The reviewers have discussed the reviews with one another and the Reviewing Editor has drafted this decision to help you prepare a revised submission.

Summary:

Curwin et al. have revised their manuscript describing a Vps4-independent requirement for ESCRT-III in the maturation of CUPS in yeast. The cover letter by Dr. Malhotra lists the changes in the revision, including the elimination of Mup1 trafficking data that conflicted with results published previously by another lab. A substantial addition to the revised manuscript is immuno EM, which supports a model in which the Acb1 cargo protein localizes to CUPS at a late stage during the maturation of this compartment. Only images are provided, however; a quantitative analysis of this pattern of localization seems not to have been performed. The timing of cargo deposition is relatively important to the authors' model that Snf7/ESCRT-III function is required for CUPS maturation – the envelopment of CUPS tubules and vesicles by the membranous saccule. The notion inferred by this reviewer is that Snf7/ESCRT-III function is needed for the saccule to capture cargo together with the immature CUPS membrane structures (a process, by the way, conceptually the same as Snf7/ESCRT-III assembly at endosomes serving to capture transmembrane protein cargoes at the site of ILV budding, first described by Teis et al., 2008 [10.1016/j.devcel.2008.08.013]). The model depicted in Figure 9 is also not wholly supported because the immuno EM experiments did not include snf7-deleted cells: the expectation from the model is that no cargo (Acb1) localization with immature CUPS membranes will be seen. In the same vein, given that the authors go to such length pointing out that CUPS maturation does not require Vps4, it seems they would have shown EM of mature CUPS in vps4-deleted cells to confirm their point beyond the localization of marker proteins using light microscopy. Indeed, in this light, the title of their manuscript seems not fully supported.

Also new to the revised manuscript is an analysis of the genetic requirements for Snf7-RFP localization during starvation in comparison with its localization during vegetative growth. The shift in Snf7-RFP in WT cells from vacuolar membranes (during growth) to puncta (during starvation) is obvious and provides a compelling case that starvation mobilizes Snf7 for other means. The Snf7-RFP-positive puncta (in wild-type cells that have been starved) are not positive for Grh1-GFP, which would seem to undermine the model that starvation repurposes Snf7 toward CUPS maturation, but Snf7-Grh1 is shown by the authors to occur transiently in Figure 4 (which is carried over from the previous version of the manuscript).

A criticism from the previous round of review that is not resolved in the revised manuscript is the nature of Snf7/ESCRT-III function required for the unconventional protein secretion pathway. To be fair, however, the nature of Snf7/ESCRT-III function is poorly understood in all pathways in which it is involved. The new data in the revised manuscript comes closer to addressing the earlier criticism that it was unclear whether Snf7 was required directly or indirectly for unconventional protein secretion. While a 'smoking gun' result is not quite in hand, all of the genetic analyses support a direct role, and the EM studies now focus on CUPS maturation, a role conceptually the same as Snf7/ESCRT-III being required for endosomal maturation.

*Reviewer 1:*

Minor points:

1) Discrepancies still exist between the text and figures and are too numerous to itemize. For example, Figure 6 shows time points that are different than those stated in the text.

2) Figure 8 (the immuno EM data) would benefit from a 'label legend' in the figure itself which states that 15-nm gold = Grh1-GFP and 10-nm gold = Acb1.

3) At the end of the subsection “The role of ESCRTS in unconventional secretion” it states, "The pathway of Acb1 secretion is the first process that requires ESCRT-III but is independent of Vps4." The authors acknowledge in the Discussion, however, that the same situation appears to occur with pH signaling, so the sentence quoted above seems misleading.

4) The legend depicted next to the bar graph in Figure 7 needs more explanation. In particular, it would help if "saccule + CUPS" is clarified in some manner. One idea would be to label these in the tomographic models shown in the right column of panels in Figure 7. The formal legend to Figure 7 do not provide much explanation.

5) The sentence “In fact, Snf7 foci had not completely formed as later in starvation, while in *vps4Δ* cells Snf7 localized almost exclusively to such foci (Figure 6)” makes no sense to me, probably some grammatical or syntax problem.

*Reviewer #2:*

The authors have made significant improvements on the current study. The reviewer thinks that the current story is complete and suitable for publication in *eLife*.

*Reviewer #3:*

Without reiterating previous summaries etc. – I feel that the authors have clearly delineated the pathway they are studying and defined the role of the ESCRT pathway proteins in the biogenesis of CUPS and secretion of Acb1. The description and conclusions are now well balanced and contribute new insight into this unconventional secretory pathway. I recommend acceptance more or less as is.

[Editors’ note: a previous version of this study was rejected after peer review, but the authors submitted for reconsideration. The previous decision letter after peer review is shown below.]

Thank you for submitting your work entitled "Vps4 independent function of ESCRTs in maturation of CUPS for unconventional secretion of Acb1" for consideration by *eLife*. Your article has been reviewed by three peer reviewers, and the evaluation has been overseen by Randy Schekman as the Senior Editor and Reviewing Editor. Our decision has been reached after consultation between the reviewers. Based on these discussions and the individual reviews below, we regret to inform you that your work will not be considered further for publication in *eLife*.

The following individual involved in the review of your submission has agreed to reveal his identity: Greg Odorizzi (peer reviewer).

As you will see from the detailed comments below, there was agreement that the results are inconclusive and that much remains before this pathway could be understood in some meaningful way. Reviewer #3 makes the intriguing comparison between your results and those reported previously for a pathway of pH sensing in fungi.

*Reviewer #1:*

The molecular mechanisms for unconventional secretion are largely unknown. Acb1 is one of the intensely studied cargoes undergoing unconventional secretion in yeast. Previous studies done by the Malhotra lab indicated a CUPS structure positive for Grh1 essential for the secretion of Acb1. In addition, molecular machineries related to MVB, such as ESCRTs, and autophagy are also involved in the release of Acb1. The genuine requirement of these factors for the secretion of Acb1 is uncertain because the previous approach relied on a functional test of the active processed Acb1. A deficiency of Acb1 secretion in this assay is direct and may be caused be the defect of Acb1 processing instead of secretion. In the current work, the Malhotra lab optimized a previous protocol to partially open the cell wall of yeast to release the secreted Acb1 trapped between the plasma membrane and cell wall. Although this approach is limited to certain applications as indicated in the manuscript, the authors determined a direct requirement of ESCRTs in the secretion of Acb1. In contrast to the MVB pathway, the authors found that a different requirement of ESCRTs i.e. ESCRTs I-III, but not 0, are required for Acb1 secretion. Therefore, the involvement of ESCRTs could be a novel mechanism distinct from canonical one for MVB pathway. The authors then employed imaging studies to analyze the CUPS structure and found a dynamic relationship between Grh1 and SNF7 essential for the maintenance of CUPS. In summary the study further implies the involvement of ERSCRTs in the secretion of Acb1. The key point of the manuscript is an unconventional requirement of ESCRT complexes in unconventional secretion. However, most of the data is supporting this point is largely descriptive and the mechanistic insight is not sufficient. The author should consider deepening the mechanistic insight and enhancing the impact of the current data through the following points.

1) It is hard to know if SNF7 could represent all other ESCRTs. The author should consider imaging the other ESCRTs to validate their conclusion. It is important to define the hierarchical relationship between these ESCRT complexes in Acb1 secretion or the contact between their compartments and Grh1 compartment considering they may act differently than the canonical manner.

2) Although the authors described a dynamic relationship between Grh1 and SNF7, how this proposed contact and coalesce transport Acb1 is unclear. The author should consider determining if Acb1 resides in these compartments and if the dynamic contact could lead to the transfer of Acb1 from Grh1 to SNF7 or other ESCRT compartments. A negative control of ESCRT-0 should also be considered to strengthen the current conclusion.

3) The authors should consider extending their CLEM to determine the structure of SNF7 and the hybrid compartment of Grh1 and SNF7 to clarify how these two compartments cooperates in Acb1 secretion. Other ESCRT compartment may also considered visualizing with this approach.

4) The authors showed the ESCRT-0 is not required for the resorption of CUPS. Are the other ESCRTs required for the resorption of the CUPS? Could depletion of the ESCRTs increase the rate of CUPS resorption?

*Reviewer #2:*

Curwin et al. describe a method to identify yeast genes that are required for the unconventional protein secretion pathway followed by Acb1, and they use this method to determine that Acb1 secretion requires genes encoding protein subunits of ESCRT-I, -II, and -III. Not among the required genes is VPS4, which encodes the ATPase that mediates ESCRT-III disassembly. Because Vps4 and ESCRT-III are required in unison for all other described ESCRT-dependent processes (MVB sorting, cytokinesis, viral budding, and other events less well characterized), the authors contend that their study describes a novel case in which ESCRT-III functions independently of Vps4 regulation. However, this function is not determined in their manuscript. What the experiments show, instead, is a requirement for Snf7 (the principle subunit of ESCRT-III) at some stage during the biogenesis of CUPS, the membranous structure that arises when unconventional protein secretion of Acb1 is induced by starvation of yeast cells. Although how Snf7 is involved (directly or indirectly) in CUPS biogenesis is not determined, tomography of CUPS elements located in cells by CLEM suggests to the authors that Snf7 is required for maturation of nascent CUPS because cells deleted for SNF7 lack the membranous cisternae that wrap CUPS vesicles. Time-lapse fluorescence microscopy are consistent with such a role for Snf7 because it transiently colocalizes with a CUPS marker (Grh1) upon starvation-induced Acb1 secretion.

Succinctly, Curwin et al. show evidence for Vps4-independent ESCRT function in the unconventional protein secretion pathway, but the nature of this function is not determined. For instance, it is unclear if ESCRTs have a direct role in this pathway. By contrast, studies describing 'new' roles for ESCRTs outside the big three (MVB sorting, cytokinesis, and viral budding) have shown interactions between ESCRTs and components involved in nuclear pore quality control (Webster et al., 2014, doi:10.1016/j.cell.2014.09.012), post-mitotic nuclear envelope reformation (Vietri et al., 2015, doi:10.1038/nature14408), exosome biogenesis ((Baietti et al., 2012, doi:10.1038/ncb2502), microvesicle shedding (Nabhan et al., 2012, do:10.1073/pnas.1200448109), and plasma membrane wound repair (Jimenez et al., 2014, doi:10.1126/science.1247136).

Given the ill-defined nature of ESCRT involvement in the unconventional protein secretion pathway described by Curwin et al., the authors make two statements that seem premature. First, "these findings prompt us to propose the involvement of a novel AAA-ATPase in the recycling of Snf7 from the membranes during Acb1 secretion." Without knowing if Snf7/ESCRT-III has a direct role in their pathway, I find it difficult to propose a novel ATPase being involved. The second statement that is difficult to support without knowing if ESCRTs have a direct role is the authors' conclusion "that ESCRT-I likely functions to recruit Acb1 or an accessory component to the Snf7 compartment or CUPS without affecting their morphology." This supposition is based on light microscopy showing CUPS biogenesis is unaffected in the absence of ESCRT-I, and is interpreted in the light of ESCRT-I having a role in cargo recognition during MVB sorting.

Figure 3 shows that the authors' conditions of starvation blocks endocytosis of Mup1-GFP, yet the converse result was shown by Jones et al., 2012 (doi:10.1111/j.1600-0854.2011.01314), wherein leucine deprivation upregulated turnover of plasma membrane proteins (including Mup1) via endocytosis and MVB sorting to the vacuole. It seems Curwin et al. need to reconcile this discrepancy, given that their results contradict the primary conclusion of the Jones study.

The Mup1-GFP results described above leads the authors to conclude, "Altogether, these data explain the surprising lack of Vps4 involvement in ESCRT III dependent Acb1 secretory pathway." How? To me, the finding that Mup1-GFP is not subjected to MVB sorting upon starvation (again, contradicting the Jones study) does not explain why unconventional secretion of Acb1 occurs independently of Vps4.

*Reviewer #3:*

This manuscript describes the development of a convincing assay for monitoring unconventional secretion of Acb1 from yeast and then uses it to extensively probe the role of the ESCRT machinery in this process. The resulting experiments elegantly define which of the ESCRT pathway components are required for this unconventional secretion (ESCRT-0, Vps4, and the Vps4 regulatory ESCRT-III like proteins are not; ESCRT-I, ESCRT-II, and core ESCRT-III proteins are). Additional experiments looking at organelle morphology (the CUPS and adjacent Snf7-positive membranes) are presented as part of the effort to understand what this seemingly novel and non-canonical combination of ESCRT requirements reveals about the pathway responsible for unconventional secretion. The authors conclude that their studies convincingly rule out roles for "ordinary" MVB-like organelles in unconventional secretion and suggest that novel yet unclear roles for the required components of the ESCRT machinery must be at play. Overall, the clear establishment of pathway requirements represents an important step forward that should be of significant general interest.

There is however one important set of related data that the authors should consider in thinking about the implications of their results, the pathway responsible for sensing extracellular pH signals in fungi. This pathway is dependent on many of the same ESCRT components reported here, and importantly also does not depend on Vps4. Key papers that suggest potential similarities between the data reported in the present manuscript and this pathway include:

"Liaison alkaline: Pals entice non-endosomal ESCRTs to the plasma membrane for pH signaling." Penalva et al., Curr. Opin. Microbiol. 2014 PMID 25460796

(This review summarizes and references much of the relevant earlier literature.)

"Signaling events of the Rim101 pathway occur at the plasma membrane in a ubiquitination-dependent manner." Obara and Kihara, Mol. Cell. Biol. 2014 PMID 25002535

"The Cryptococcus neoformans alkaline response pathway: identification of a novel Rim pathway activator" PLoS Genetics 2015 PMID 25859664

"Analysis of the dual function of the ESCRT-III protein Snf7 in endocytic trafficking and in gene expression" Biochem J 2009 PMID 19725809

---

## [Author Response]

Reviewer 1:

Minor points:

1) Discrepancies still exist between the text and figures and are too numerous to itemize. For example, Figure 6 shows time points that are different than those stated in the text.

Done. Changes also made in description of Figure 4 and Figure 6 to clarify the confusion.

2) Figure 8 (the immuno EM data) would benefit from a 'label legend' in the figure itself which states that 15-nm gold = Grh1-GFP and 10-nm gold = Acb1.

Done.

3) At the end of the subsection “The role of ESCRTS in unconventional secretion” it states, "The pathway of Acb1 secretion is the first process that requires ESCRT-III but is independent of Vps4." The authors acknowledge in the Discussion, however, that the same situation appears to occur with pH signaling, so the sentence quoted above seems misleading.

We have changed this sentence and in fact edited this statement throughout the text. For example,

In Introduction

Changed from: These results indicate for the first time a Vps4 independent role of ESCRT-III in membrane remodeling.

To:These results indicate a Vps4 independent role of ESCRT-III in membrane remodeling.

In Results

Subheading: ESCRT-I, II and III are required for Acb1 secretion

Last paragraph changed from:These results suggest that Acb1 secretion requires ESCRT-III function but not Vps4 activity. This is an unexpected finding since all other known ESCRT dependent processes require ESCRT-III and Vps4.

To: These results suggest that Acb1 secretion requires ESCRT-III function but not Vps4 activity.

Subheading: Snf7 recruitment to CUPS and Acb1 secretion are accelerated in vps4Δ cells

First paragraph changed significantly to clarify confusion.

In Discussion

Subheading: The role of ESCRTS in unconventional secretion

Last line changed from:The pathway of Acb1 secretion is the first process that requires ESCRT-III but is independent of Vps4.

To: However, the pathway of Acb1 secretion that requires ESCRT-III is independent of Vps4.

4) The legend depicted next to the bar graph in Figure 7 needs more explanation. In particular, it would help if "saccule + CUPS" is clarified in some manner. One idea would be to label these in the tomographic models shown in the right column of panels in Figure 7. The formal legend to Figure 7 do not provide much explanation.

The legend and figure have been modified.

*5) The sentence “In fact, Snf7 foci had not completely formed as later in starvation, while in vps4Δ cells Snf7 localized almost exclusively to such foci (Figure 6)” makes no sense to me, probably some grammatical or syntax problem.*

This is the description of Figure 6, already addressed above.

[Editors’ note: the author responses to the previous round of peer review follow.]

Reviewer #1:

1) It is hard to know if SNF7 could represent all other ESCRTs. The author should consider imaging the other ESCRTs to validate their conclusion.

Snf7 is transiently recruited to the Grh1 containing membranes, which makes its localization very difficult. We have now localized Snf7 in starvation conditions in the absence of individual ESCRTS. We find that recruitment of Snf7 to membranes during starvation follows the same principle as for MVB pathway (Figure 5). ESCRT-I and II are mainly cytosolic under normal conditions and expressed at very low levels compared to ESCRT-III. A faint MVB localization can be observed and this location was largely unchanged upon starvation. We didn’t include this, as we feel this does not add anything to the paper.

It is important to define the hierarchical relationship between these ESCRT complexes in Acb1 secretion or the contact between their compartments and Grh1 compartment considering they may act differently than the canonical manner.

We have addressed this by localizing Snf7 in all ESCRT deleted strains (Figure 5). Moreover, we go on to show accelerating Snf7 recruitment to CUPS by deleting Vps4 correlates with accelerated Acb1 secretion. Additionally, we show Acb1 secretion requires the ability of Snf7 to homooligomerize.

2) Although the authors described a dynamic relationship between Grh1 and SNF7, how this proposed contact and coalesce transport Acb1 is unclear.

We now include new data from immuno-EM and CLEM analysis that this contact results in the production of stable CUPS that contains Acb1 (Figure 7, Figure 8 and Figure 9).

The authors should consider determining if Acb1 resides in these compartments and if the dynamic contact could lead to the transfer of Acb1 from Grh1 to SNF7 or other ESCRT compartments. A negative control of ESCRT-0 should also be considered to strengthen the current conclusion.

Done (Figure 8). ESCRT-0 has no role in our pathway and we have not tested this by immuno-EM analysis as we expect that it would have no effect. These are extremely time consuming experiments and we sincerely hope that the reviewer would appreciate the effort and the returns. With all due respect, this experiment does not add anything to our proposal. It would be the same as knocking down any other protein that does not have a role in our pathway.

3) The author should consider extending their CLEM to determine the structure of SNF7 and the hybrid compartment of Grh1 and SNF7 to clarify how these two compartments cooperates in Acb1 secretion.

This is difficult as the co-localization is very transient. We have tried but we have not been able to capture this event. Had we the data, we would have included it in the paper. We are trying to identify the Snf7 containing membranes that engulf Grh1 containing vesicles. Could this not be an issue for a later paper? Must we identify and present every aspect of this pathway in one paper? Isn’t showing the immune-EM data on the organization of CUPS at different stages, and in Snf7 dependent reaction, more than sufficient and novel?

Other ESCRT compartment may also considered visualizing with this approach.

This is again an add-on at huge expense without any significant addition to our proposal. The findings that ESCRT III recruitment to the CUPS is similar to the MVB pathway makes this request less important for presentation in our paper. ESCRTS I and II are mostly cytoplasmic, as we have stated.

*4) The authors showed the ESCRT-0 is not required for the resorption of CUPS. Are the other ESCRTs required for the resorption of the CUPS? Could depletion of the ESCRTs increase the rate of CUPS resorption?* The reviewer must mean Vps4 and not ESCRT-0 in this case. We have reabsorption data for all ESCRTs but we did not include it to prevent clutter. The answer is no, none of the ESCRTS are required for the reabsorption of CUPS upon switching from potassium acetate to growth medium. This is now included in a new figure (Figure 3, we show the example of Vps4 and state the lack of involvement of the other as data not shown).

Reviewer #2:

Although how Snf7 is involved (directly or indirectly) in CUPS biogenesis is not determined, tomography of CUPS elements located in cells by CLEM suggests to the authors that Snf7 is required for maturation of nascent CUPS because cells deleted for SNF7 lack the membranous cisternae that wrap CUPS vesicles.

The reviewer is correct and now our new data shows that Snf7 allows the engulfment of Grh1 contains membranes to generate a new stable compartment that contains Acb1. This is now in the paper.

Time-lapse fluorescence microscopy are consistent with such a role for Snf7 because it transiently colocalizes with a CUPS marker (Grh1) upon starvation-induced Acb1 secretion.

Yes. Thanks.

*Succinctly, Curwin* et al. *show evidence for Vps4-independent ESCRT function in the unconventional protein secretion pathway, but the nature of this function is not determined. For instance, it is unclear if ESCRTs have a direct role in this pathway. By contrast, studies describing 'new' roles for ESCRTs outside the big three (MVB sorting, cytokinesis, and viral budding) have shown interactions between ESCRTs and components involved in nuclear pore quality control (Webster et al., 2014, doi:10.1016/j.cell.2014.09.012), post-mitotic nuclear envelope reformation (Vietri et al., 2015, doi:10.1038/nature14408), exosome biogenesis ((Baietti et al., 2012, doi:10.1038/ncb2502), microvesicle shedding (Nabhan et al., 2012, do:10.1073/pnas.1200448109), and plasma membrane wound repair (Jimenez et al., 2014, doi:10.1126/science.1247136).*

We have demonstrated Snf7 is required for Acb1 secretion and CUPS morphology and that a pool of Snf7 localizes transiently to CUPS. The reviewer has even agreed our time-lapse microscopy data show this. We are uncertain what further evidence might satisfy this reviewer to show a direct involvement of ESCRTs in our pathway.

Looking in detail at the data presented in the references given above, Vietri et al. (Nature 2015) showed similar evidence to reach their conclusions. Some ESCRT components were localized around chromatin, knock-down led to various phenotypes such as dysfunction in nuclear envelope re-sealing and spindle disassembly and CLEM based analysis. Similarly, Baietti et al. (NCB 2012) simply examined exosome production and/or content as a read out and presented one fluorescence microscopy experiment which was based on a dysfunctional endosome. Nabbhan et al. (PNAS 2012), Jimenez et al. (Science 2012) and Webster et al. (Cell 2014) do go further in defining the mechanism of ESCRT function in the various processes described, showing direct protein interactions. However, we do not propose that Snf7 or ESCRTs must interact physically with Grh1, only that the compartments interact. The localization of Snf7 in various ESCRT mutants now shown has clarified the function of ESCRTs in our pathway.

*Given the ill-defined nature of ESCRT involvement in the unconventional protein secretion pathway described by Curwin* et al.*, the authors make two statements that seem premature. First, "these findings prompt us to propose the involvement of a novel AAA-ATPase in the recycling of Snf7 from the membranes during Acb1 secretion." Without knowing if Snf7/ESCRT-III has a direct role in their pathway, I find it difficult to propose a novel ATPase being involved.*

We make no comment on the mechanism of Snf7 disassembly, but it is clearly independent of Vps4. The identification of this new mechanism of Snf7 disassembly is beyond the scope of this paper.

The second statement that is difficult to support without knowing if ESCRTs have a direct role is the authors' conclusion "that ESCRT-I likely functions to recruit Acb1 or an accessory component to the Snf7 compartment or CUPS without affecting their morphology." This supposition is based on light microscopy showing CUPS biogenesis is unaffected in the absence of ESCRT-I, and is interpreted in the light of ESCRT-I having a role in cargo recognition during MVB sorting.

This was just a statement in the Discussion and we have removed it from the text.

Figure 3 shows that the authors' conditions of starvation blocks endocytosis of Mup1-GFP, yet the converse result was shown by Jones et al., 2012 (doi:10.1111/j.1600-0854.2011.01314), wherein leucine deprivation upregulated turnover of plasma membrane proteins (including Mup1) via endocytosis and MVB sorting to the vacuole. It seems Curwin et al. need to reconcile this discrepancy, given that their results contradict the primary conclusion of the Jones study.

This figure has been removed. We now state quite clearly that the compartment CUPS is distinct from a standard MVB and therefore there is no need to state whether the MVB pathway is functional or not under our experimental conditions. The paper of Babst and colleagues (Jones et al., 2012) and also Teis and colleagues (Muller et al. 2015 *eLife*) describes the process of endocytosis and MVB function in cells that are depleted of amino acids or treated with Rapamycin. Our paper describes these processes in cells cultivated in potassium acetate. These are very different conditions, with the most obvious difference being the presence of glucose in the aforementioned studies, but not in ours. Moreover, we have already shown that CUPS formation is independent of Rapamycin treatment (Bruns et al., 2011), and although not published we have data that Acb1 secretion, similarly, is not triggered by Rapamycin treatment. So there is no issue with regards to a contradiction of the published work of Babst and colleagues.

The Mup1-GFP results described above leads the authors to conclude, "Altogether, these data explain the surprising lack of Vps4 involvement in ESCRT III dependent Acb1 secretory pathway." How? To me, the finding that Mup1-GFP is not subjected to MVB sorting upon starvation (again, contradicting the Jones study) does not explain why unconventional secretion of Acb1 occurs independently of Vps4.

Our new data clarifies this further by showing a lack of Vps4 in the secretion of Acb1. We have explained this clearly in the new text.

Reviewer #3:

There is however one important set of related data that the authors should consider in thinking about the implications of their results, the pathway responsible for sensing extracellular pH signals in fungi.

[…]

"Analysis of the dual function of the ESCRT-III protein Snf7 in endocytic trafficking and in gene expression" Biochem J 2009 PMID 19725809

We have genetic data that the pH sensing pathway and CUPS/Acb1 secretion require different set of ESCRT genes. There are some similarities that we now acknowledge and discuss in the text. We also state the possibility of signaling mechanisms shared by these two diverse cellular responses.